# 5-HT regulates resistance to aumolertinib by attenuating ferroptosis in lung adenocarcinoma

Yuanying Feng [ID][1,2,3,4,7], Yuchao He[1,3,7], Ran Zuo [ID][3,5,7], Wenchen Gong [ID][3,6], Yuan Gao[1,2,3], Yun Wang[1,2,3], Yu Wang[1,3], Wenshuai Chen[1,3], Liwei Chen [ID][1,3], Yi Luo[1,3], Dongqi Yuan[1,2,3], Peng Chen [ID][2,3✉] & Hua Guo [ID][1,3✉]

## Abstract

**Resistance to epidermal growth factor receptor (EGFR)-tyrosine kinase inhibitor (TKI) remains a critical clinical challenge in EGFR mutant lung adenocarcinoma (LUAD). Therefore, it is urgent to explore personalized treatment strategies based on distinct resistance mechanisms to reverse EGFR-TKI resistance. Herein, we found that HER2 S310F mutation contributes to third-generation EGFR-TKI resistance, driven by the accumulation of neurotransmitter 5-hydroxytryptamine (5-HT). Mechanistically, 5-HT interacted with 5-HT3 receptor, triggering calcium ion ($Ca^{2+}$) influx and subsequent activation of the $Ca^{2+}$/CAMKK2/AMPK pathway. This pathway activation conferred ferroptosis resistance, thereby driving aumolertinib resistance. 5-HT3 receptor (HTR3) antagonists were pinpointed as potential agents for reversing aumolertinib resistance through drug library screening and transcriptomics analysis. We demonstrated that pharmacologically targeting 5-HT/HTR3 signaling with the clinically approved HTR3 antagonist palonosetron effectively restores aumolertinib sensitivity. Importantly, we showed that elevated 5-HT levels in patient plasma play a potential role in predicting EGFR-TKI resistance. Our data highlight the critical role of 5-HT and ferroptosis in the development of aumolertinib resistance, and propose HTR3 antagonists as a novel combination therapy strategy for LUAD treatment with aumolertinib.**

**Keywords** Aumolertinib; EGFR-TKI Resistance; Ferroptosis; HTR3 Antagonists; 5-Hydroxytryptamine
**Subject Categories** Cancer; Respiratory System

## Introduction

In the global cancer statistics of 2022, lung cancer ranked first in cancer incidence and mortality rates (Bray et al, 2024). LUAD is the most common histological subtype. LUAD patients with activating mutations in the EGFR clearly benefit from EGFR-TKI therapy (Pao and Girard, 2011). However, the responsiveness to EGFR-TKI therapy varies among patients (Kobayashi et al, 2005). In addition to the modifications in targeted genes, cancer cells can circumvent the inhibited signaling pathways by activating alternative tyrosine kinase receptors that facilitate cell survival and proliferation (Wu and Shih, 2018). For instance, MET or HER2 amplification frequently represents an EGFR-independent mechanism of resistance to EGFR-TKI (Schmid et al, 2020; Ferro et al, 2024). Furthermore, recent investigations have revealed that certain HER2 mutations, such as exon 16 skipping mutations, are present in patients exhibiting resistance to EGFR-TKI (Hsu et al, 2020). The correlation between other HER2 mutations and EGFR-TKI sensitivity remains to be elucidated.

An increasing number of studies have reported the role of neurotransmitters in tumorigenesis, development, and metastasis, emphasizing the connection between neuroendocrine and tumor cell biology (Venkataramani et al, 2019; Zeng et al, 2019). 5-HT is recognized as the most versatile neurotransmitter, playing crucial roles in the nervous system, gastrointestinal system, and energy homeostasis (Yabut et al, 2019). In recent years, increasing evidence demonstrated that 5-HT can facilitate tumorigenesis and tumor progression across various malignancies (Jiang et al, 2017; Tone et al, 2020; Liu et al, 2023; Tu et al, 2023). 5-HT exerts its effects through the activation of 5-HT receptors (HTRs), which are categorized into seven types, ranging from HTR1 to HTR7. All HTRs are G protein-coupled receptors (GPCRs), with the exception of HTR3, which functions as a ligand-gated ion channel (McCorvy and Roth, 2015). Notably, HTR3 has been reported to be essential for cell viability and proliferation in LUAD (Tone et al, 2020; Wu et al, 2024). HTR3 antagonists, commonly referred to as setrons, have been utilized clinically to manage nausea and vomiting induced by chemotherapy, radiation, and anesthesia, as well as in the treatment of irritable bowel syndrome (Juza et al, 2020).

[1]Department of Tumor Cell Biology, Tianjin Medical University Cancer Institute and Hospital, 300060 Tianjin, China. [2]Department of Thoracic Oncology, Lung Cancer Diagnosis and Treatment Center, Tianjin Medical University Cancer Institute and Hospital, 300060 Tianjin, China. [3]National Clinical Research Center for Cancer, State Key Laboratory of Druggability Evaluation and Systematic Translational Medicine, Tianjin's Clinical Research Center for Cancer, Key Laboratory of Cancer Prevention and Therapy, 300060 Tianjin, China. [4]Beijing Chest Hospital, Capital Medical University, Beijing Tuberculosis and Thoracic Tumor Research Institute, 101149 Beijing, China. [5]Department of Integrative Oncology, Tianjin Medical University Cancer Institute and Hospital, 300060 Tianjin, China. [6]Department of Pathology, Tianjin Medical University Cancer Institute and Hospital, 300060 Tianjin, China. [7]These authors contributed equally: Yuanying Feng, Yuchao He, Ran Zuo. ✉E-mail: chenpeng@tjmuch.com; guohua@tjmuch.com

However, the implications of HTR3 antagonists in the development of LUAD and resistance to EGFR-TKI remain to be determined.

Ferroptosis is a newly identified programmed cell death that mainly relies on iron-mediated oxidative damage (Jiang et al, 2021). Ferroptosis has also been proven to correlate with cancer therapy resistance, and inducing ferroptosis has been demonstrated to reverse drug resistance (Zhang et al, 2022). In addition to iron ions, $Ca^{2+}$ plays a key role in the regulation of ferroptosis (Iwabu et al, 2010; Wu et al, 2024). Numerous proteins involved in cellular antioxidant defense and reactive oxygen species (ROS) production are dependent on $Ca^{2+}$, and metabolic dysregulation of $Ca^{2+}$ within cells can induce ferroptosis through interactions with iron and crosstalk between the endoplasmic reticulum and mitochondria (Sun et al, 2023). $Ca^{2+}$ homeostasis is an important driving factor in the occurrence and progression of cancer, and it affects the treatment response of cancer patients (Marchi et al, 2020).

In this study, we identified and validated that the HER2 S310F mutation contributes to third-generation EGFR-TKI resistance. Through high-throughput screening (HTS) of an FDA-approved drug library and transcriptome sequencing analysis, we pinpointed HTR3 antagonists as potential agents for reversing aumolertinib resistance. Furthermore, we demonstrated the important role of 5-HT in aumolertinib resistance. Specifically, the accumulation of 5-HT promotes resistance to ferroptosis, partially mediated by the $Ca^{2+}$/CAMKK2/AMPK signaling pathway, which contributes to aumolertinib resistance. From a therapeutic perspective, the combination of aumolertinib with HTR3 antagonists effectively overcame aumolertinib resistance, highlighting a promising combinatorial therapeutic strategy to address the emergence of aumolertinib resistance.

## Results

### HER2 S310F mutation contributes to third-generation EGFR-TKI resistance in LUAD cells

To further understand the relationship between HER2 alterations and EGFR-TKI resistance, a total of 81 patients with HER2 changes were collected from Tianjin Medical University Cancer Institute and Hospital from September 2021 to September 2023 (Table EV1). Within this cohort, 57 patients presented with HER2 mutations, whereas nine exhibited concomitant EGFR co-mutations. Notably, six instances of co-mutation were detected in patients carrying the HER2 S310F mutation, with a total of seven patients identified as harboring the HER2 S310F mutation. Among these six patients, four received treatment with third-generation EGFR-TKI targeted therapies, with the maximum duration of treatment not exceeding 7 months, and all of whom experienced disease progression. We presented the diagnostic and treatment processes of one representative case: a 75-year-old male diagnosed with stage IV NSCLC harboring both HER2 S310F and EGFR mutations. The patient exhibited disease progression after more than 5 months of targeted treatment with aumolertinib (Fig. 1A). The progression-free survival (PFS) of these patients was significantly shorter than the median PFS of 19.3 months reported in the AENEAS phase III clinical trial results (Lu et al, 2022). Thus, we speculated whether the HER2 S310F mutation contributes to third-generation EGFR-TKI resistance.

To evaluate HER2 expression in LUAD cell lines, we performed western blot analysis to quantify HER2 protein levels, utilizing the HER2-positive breast cancer cell lines SK-BR-3 and BT-474 as positive controls. Our results indicated that HER2 expression was undetectable at the protein level in EGFR-mutated LUAD cell lines H1975, PC9, and H3255 (Appendix Fig. S1A). Consequently, stable lines expressing the HER2 S310F mutant were established using these three cell lines (Fig. 1B; Appendix Fig. S1B). We conducted an in-depth evaluation of the impact of the HER2 S310F mutation in LUAD cells on their responsiveness to third-generation EGFR-TKI. Following HER2 S310F mutation, the half-maximal inhibitory concentration ($IC_{50}$) of aumolertinib in H1975, PC9 and H3255 cells significantly increased (Fig. 1C–E). We employed another third-generation EGFR-TKI, osimertinib, and observed similar results (Appendix Fig. S1C–E). To further elucidate the relationship of HER2 S310F mutation and aumolertinib resistance in vivo, we subcutaneously inoculated PC9 EV and PC9 S310F cells into BALB/c nude mice. When the tumor size reached an average of 150–200 mm³, the mice were randomly divided into four groups: EV control, EV aumolertinib, S310F control, and S310F aumolertinib. As shown in Fig. 1F-H, aumolertinib effectively inhibited PC9 EV xenograft tumors, but it did not affect PC9 S310F tumors. The body weights of the four groups of mice showed no significant differences, indicating that the toxicity of aumolertinib was manageable (Fig. 1I). Collectively, these findings suggested that HER2 S310F conferred resistance to third-generation EGFR-TKI in vitro and in vivo without obvious toxicity.

### Identification of HTR3 antagonists as sensitizers for aumolertinib

To identify pharmacological agents capable of overcoming third-generation EGFR-TKI resistance associated with the HER2 S310F mutation, we initially assessed HER2-TKI pyrotinib and the HER2 antibody-drug conjugate DS-8201. These results demonstrated that pyrotinib and DS-8201 did not exhibit a significant synergistic effect when combined with aumolertinib in H1975 S310F cells (Fig. EV1A,B). To explore potential drugs that may overcome aumolertinib resistance in HER2 S310F mutant LUAD, we conducted HTS utilizing a drug library comprising 2972 FDA-approved compounds in H1975 S310F mutant cells, with paired H1975 EV cells serving as a control (Fig. 2A). This procedure identified 27 hypersensitive compounds that significantly diminished the viability of HER2 S310F mutant H1975 cells while sparing H1975 EV cells (Appendix Table S1). Subsequently, we performed RNA-seq on H1975 EV and S310F mutant cells. We predicted effective drugs for HER2 S310F mutant cells based on differential gene expression using the Connectivity Map (cMAP) database (Fig. EV1C; Appendix Table S2). By integrating the findings from the FDA-approved library and the cMAP database predictions, four distinct inhibitor classes were identified: topoisomerase inhibitors, ALK inhibitors, HTR3 antagonists, and PI3K/AKT inhibitors (Fig. 2B). KEGG analysis of the RNA-seq data revealed a correlation between HER2 S310F mutant cells and neuroactive ligand–receptor interactions (Fig. 2C). HTR3 interacts with its ligand, the neurotransmitter 5-HT, to exert its effects, falling within the category of neuroactive ligand–receptor interactions (Chen et al, 2022). Therefore, we first verified whether HTR3 antagonists can reverse aumolertinib resistance.

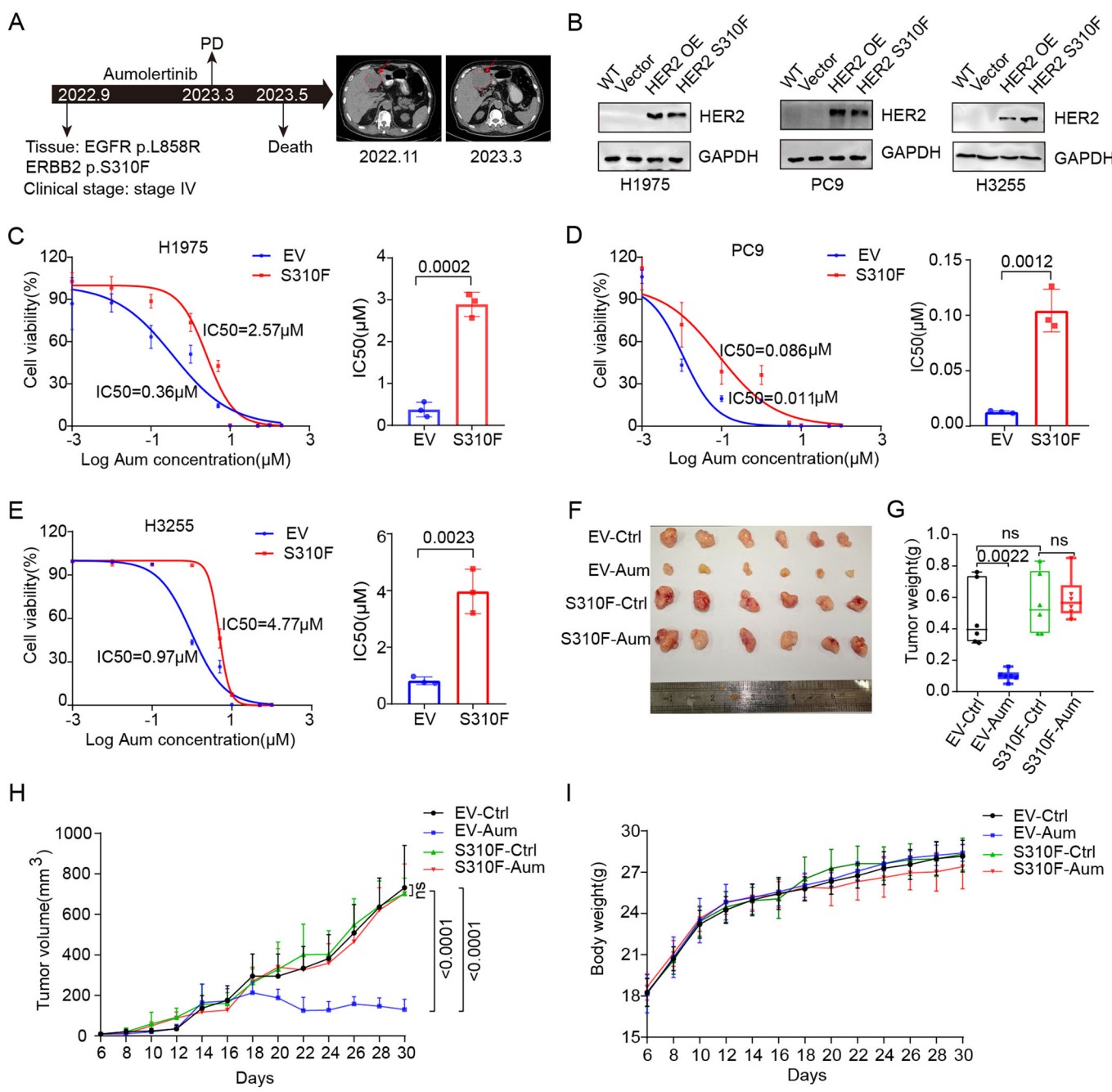

**Figure 1. HER2 S310F mutation contributes to aumolertinib resistance.**

(A) Time from diagnosis to the patient's death is indicated by the black arrow. (B) H1975, PC9 and H3255 cells stably expressing HER2 or HER2 S310F were validated by western blot. (C) $IC_{50}$ analysis of aumolertinib for 5 days by the CCK-8 assay in H1975 EV/S310F cells. Data in the right panel represents mean ± SD of three independent biological replicates. (D) $IC_{50}$ analysis of aumolertinib for 5 days by the CCK-8 assay in PC9 EV/S310F cells. Data in the right panel represents mean ± SD of three independent biological replicates. (E) $IC_{50}$ analysis of aumolertinib for 5 days by the CCK-8 assay in H3255 EV/S310F cells. Data in the right panel represents mean ± SD of three independent biological replicates. (F) Representative images of the tumors. (G) Measurement of tumor weights ($n = 6$). The boxes extend from the 25th to 75th percentile, the middle line shows the median, whiskers extend to the most extreme data. (H) Time course of tumor volumes ($n = 6$). Data are presented as mean ± SD. (I) Time course of mice weight ($n = 6$). Data are presented as mean ± SD. Two-tailed unpaired Student's $t$ test was used for statistical analysis in (C–E), one-way ANOVA was used for statistical analysis in (G, H). Source data are available online for this figure.

We utilized HTR3 antagonists from the FDA-approved library, including granisetron, tropisetron, ondansetron, dolasetron, ramosetron, and palonosetron, to assess the inhibitory effects on cell viability and combination index (CI) of HTR3 antagonists and aumolertinib in H1975 S310F mutant cells. The results indicated that all aforementioned HTR3 antagonists exhibited synergistic effects with aumolertinib (Figs. 2D and EV1D–H). Considering the high specificity and long half-life of palonosetron (Minotti, 2023),

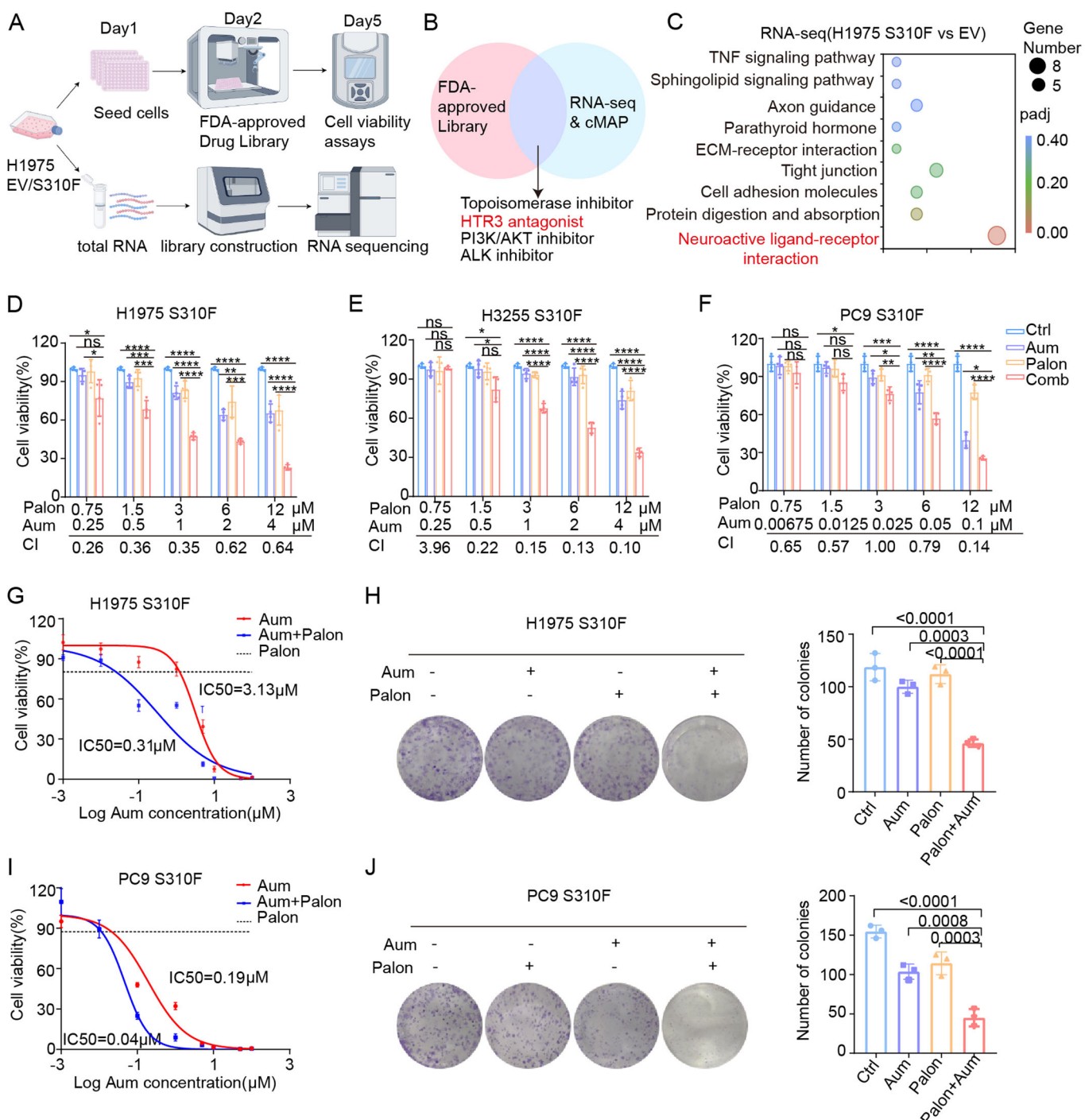

along with its significant synergistic effects when combined with aumolertinib, we selected palonosetron for further investigation. We conducted additional validation and demonstrated that palonosetron and aumolertinib exhibited substantial synergistic effects in PC9 and H3255 S310F mutant cells (Fig. 2E,F). Subsequently, we validated the effect of palonosetron on $IC_{50}$ of aumolertinib through CCK8 experiments, revealing that low-dose palonosetron significantly reduced $IC_{50}$ of aumolertinib in HER2 S310F mutant cells (Fig. 2G,I). We further evaluated the effect of

the two drugs on colony formation through plate cloning experiments. The combination of palonosetron and aumolertinib demonstrated significantly stronger inhibition of colony formation than either drug alone (Fig. 2H,J). To exclude potential off-target effects of HTR3 antagonists, we constructed stable HTR3A-knockdown cell lines in H1975 S310F and PC9 S310F cells using shRNA lentiviral transduction. Subsequent CCK-8 assays revealed that HTR3A knockdown significantly reduced aumolertinib's $IC_{50}$, indicating HTR3's essential role in 5-HT-induced resistance

**Figure 2.  Identification of HTR3 antagonists as sensitizers for aumolertinib.**

(A) The flowchart depicting the HTS of the FDA-approved drug library and RNA sequencing was illustrated by Figdraw. (B) The intersection of the FDA-approved drug library and the cMAP database screening results was shown. (C) KEGG for metabolic pathways referring to differential genes, respectively. ClusterProfiler was used for gene enrichment analysis of KEGG, and $P \leq 0.05$ was used as the enrichment cut-off. (D) Cell viability and CI analysis in H1975 S310F cells treated with indicated concentrations of aumolertinib, palonosetron alone, or in combination for 72 h ($n = 4$). P value (Aum 0.25 μM, Palon 0.75 μM): Ctrl vs Comb, $P = 0.0154$, Palon vs Comb, $P = 0.0263$; P value (Aum 0.5 μM, Palon 1.5 μM): Ctrl vs Comb, $P < 0.0001$, Aum vs Comb, $P = 0.0005$, Palon vs Comb, $P = 0.0002$; P value (Aum 1 μM, Palon 3 μM): Ctrl vs Comb, $P < 0.0001$, Aum vs Comb, $P < 0.0001$, Palon vs Comb, $P < 0.0001$; P value (Aum 2 μM, Palon 6 μM): Ctrl vs Comb, $P < 0.0001$, Aum vs Comb, $P = 0.0037$, Palon vs Comb, $P = 0.0001$; P value (Aum 4 μM, Palon 12 μM): Ctrl vs Comb, $P < 0.0001$, Aum vs Comb, $P < 0.0001$, Palon vs Comb, $P < 0.0001$. (E) Cell viability and CI analysis in H3255 S310F cells treated with indicated concentrations of aumolertinib, palonosetron alone, or in combination for 72 h ($n = 4$). P value (Aum 0.5 μM, Palon 1.5 μM): Ctrl vs Comb, $P = 0.0114$, Aum vs Comb, $P = 0.0254$; P value (Aum 1 μM, Palon 3 μM): Ctrl vs Comb, $P < 0.0001$, Aum vs Comb, $P < 0.0001$, Palon vs Comb, $P < 0.0001$; P value (Aum 2 μM, Palon 6 μM): Ctrl vs Comb, $P < 0.0001$, Aum vs Comb, $P < 0.0001$, Palon vs Comb, $P < 0.0001$; P value (Aum 4 μM, Palon 12 μM): Ctrl vs Comb, $P < 0.0001$, Aum vs Comb, $P < 0.0001$, Palon vs Comb, $P < 0.0001$. (F) Cell viability and CI analysis in PC9 S310F cells treated with indicated concentrations of aumolertinib, palonosetron alone, or in combination for 72 h ($n = 4$). P value (Aum 0.0125 μM, Palon 1.5 μM): Ctrl vs Comb, $P = 0.0114$; P value (Aum 0.025 μM, Palon 3 μM): Ctrl vs Comb, $P = 0.0002$, Aum vs Comb, $P = 0.0209$, Palon vs Comb, $P = 0.009$; P value (Aum 0.05 μM, Palon 6 μM): Ctrl vs Comb, $P < 0.0001$, Aum vs Comb, $P = 0.0032$, Palon vs Comb, $P < 0.0001$; P value (Aum 0.1 μM, Palon 12 μM): Ctrl vs Comb, $P < 0.0001$, Aum vs Comb, $P = 0.0104$, Palon vs Comb, $P < 0.0001$. (G) Dose-response curves determined by the CCK-8 assay were used to calculate the $IC_{50}$ values of aumolertinib for 5 days in the presence or absence of palonosetron in H1975 S310F cells. Black dotted lines indicate the cell viability at a concentration of 5 μM palonosetron for 5 days ($n = 4$). (H) The colony-forming efficiency of H1975 S310F was determined. These cells were treated with the indicated drugs at the same time for 10 days ($n = 3$). (I) Dose-response curves determined by the CCK-8 assay were used to calculate the $IC_{50}$ values of aumolertinib for 5 days in the presence or absence of palonosetron in PC9 S310F cells. Black dotted lines indicate the cell viability at a concentration of 5 μM palonosetron for 5 days ($n = 4$). (J) The colony-forming efficiency of PC9 S310F was determined. These cells were treated with the indicated drugs at the same time for 10 days ($n = 3$). Data are presented as mean ± SD. Statistical test: one-way ANOVA, ns not significant, *$P < 0.05$, **$P < 0.01$, ***$P < 0.001$, and ****$P < 0.0001$. Source data are available online for this figure.

(Fig. EV2A–D). Pharmacological inhibition of HTR3 similarly reduced aumolertinib's $IC_{50}$, collectively demonstrating that the enhanced drug sensitivity induced by HTR3 antagonists is directly attributable to on-target HTR3A inhibition rather than off-target effects.

## Neurotransmitter 5-HT accumulation leads to aumolertinib resistance

To further investigate the mechanism by which HTR3 antagonists reverse aumolertinib resistance conferred by the HER2 S310F mutation, we first examined whether the 5-HT3 receptor is upregulated in HER2 S310F mutant cells. The results of western blot indicated that the expression of HTR3 in HER2 S310F mutant cells of H1975, PC9, and H3255 remained unchanged compared with their control cells. (Fig. 3A–C). We further analyzed the content of 5-HT, the ligand of HTR3, in the cell culture medium, and the results showed a marked elevation in the 5-HT content within the culture medium of HER2 S310F mutant cells (Fig. 3D–F). These findings implied that elevated 5-HT levels may be linked to the development of aumolertinib resistance. To verify this hypothesis, we administered exogenous 5-HT and assessed its impact on $IC_{50}$ of aumolertinib. After exogenous supplementation of 2.5 μM 5-HT, the $IC_{50}$ of aumolertinib significantly increased (Fig. 3G–I). Exogenous supplementation of 2.5 μM 5-HT also significantly increased the $IC_{50}$ of another third-generation EGFR-TKI, osimertinib (Appendix Fig. S2A–C). The above results suggested that elevated levels of 5-HT were associated with the development of resistance to aumolertinib.

## HER2 S310F mutation induces the reduced expression of MAOA

To elucidate the mechanisms responsible for the elevated 5-HT level, we analyzed the genes involved in 5-HT biosynthesis, transport, and metabolism (Fig. 4A). Transcriptomic sequencing revealed a significant downregulation of the key enzyme MAOA,

which was corroborated by qPCR in H1975 and PC9 cells (Fig. 4B–D). Western blot analysis further confirmed these results, showing a significant decrease in MAOA protein levels in HER2 S310F mutant cells (Fig. 4E). MAOA is a flavoprotein localized in the outer mitochondrial membrane, catalyzing the oxidative deamination of monoamine neurotransmitters (Santin et al, 2021). Recent studies have increasingly highlighted the role of MAOA in the pathogenesis and progression of cancer (Wei and Wu, 2024). MAOA levels were found to be lower in various tumors relative to adjacent normal tissues (Appendix Fig. S3A). Specifically, MAOA expression was reduced in LUAD compared with normal lung tissue (Appendix Fig. S3B) and was significantly lower in EGFR-TKI-resistant cells than wild-type cells (Appendix Fig. S3C). Low MAOA expression was correlated with poor prognosis, as analyzed through the GEO database (Appendix Fig. S3D). IHC results confirmed that MAOA expression was significantly lower in patients harboring concurrent EGFR and HER2 S310F mutations compared with those with only EGFR mutation (Fig. 4F). Stable MAOA overexpression and vector cell lines were established in H1975 and PC9 S310F cells (Fig. EV3A,B). The results showed that MAOA overexpression decreased 5-HT content within the culture medium and $IC_{50}$ of aumolertinib (Fig. EV3C–F). In addition, we established stable MAOA knockdown cell lines in PC9 and H1975 cells. Compared to shCtrl group, sh-MAOA significantly increased aumolertinib's $IC_{50}$ (Fig. EV3G–J). Collectively, these results indicated that the HER2 S310F mutation induces the reduced expression of MAOA and the accumulation of 5-HT.

As a result of the significant decrease in the expression of MAOA at the RNA and protein levels, we first considered that the HER2 S310F mutation affected the transcription of MAOA. Transcriptome sequencing analysis indicated a reduction in the NF-kappa-B (NF-κB) signaling pathway in HER2 S310F mutant cells (Fig. EV4A). We utilized the JASPAR database to analyze the potential existence of three P65 binding sites in the promoter region of MAOA. We then delineated the most probable binding site sequence (Fig. EV4B), which indicated that P65 may be one of the transcription factors regulating MAOA. Our results from nuclear–cytoplasmic fractionation

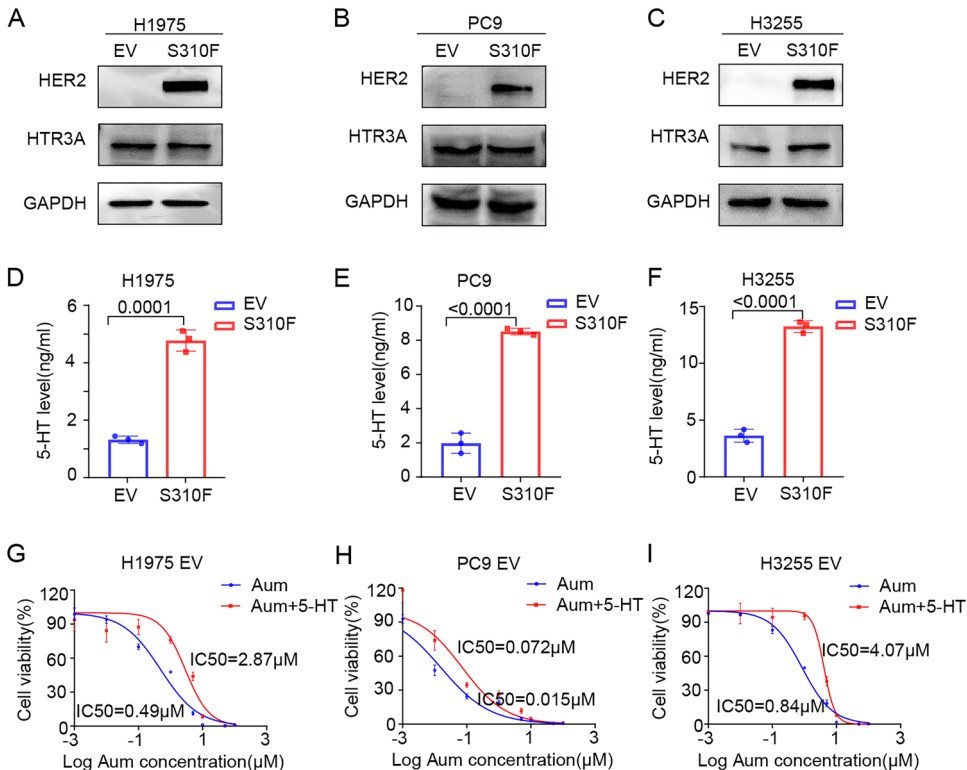

**Figure 3.  Accumulation of 5-HT mediates aumolertinib resistance.**

(A–C) Western blotting verification of HER2 and HTR3 in H1975, PC9, H3255 EV and S310F cells. (D-F) 5-HT levels in the supernatant ($n = 3$). (G–I) Dose–response curves determined by the CCK-8 assay were used to calculate the $IC_{50}$ values of aumolertinib for 5 days in the presence or absence of 2.5 µM 5-HT ($n = 4$). Data are presented as mean ± SD. Statistical test: Two-tailed unpaired Student's $t$ test. Source data are available online for this figure.

experiments indicated a reduced nuclear translocation of P65 (Fig. EV4C,D). Furthermore, immunofluorescence assays corroborated the diminished nuclear localization of P65 in H1975 and PC9 HER2 S310F mutant cells (Fig. EV4E,F).

We designed primers targeting three predicted P65 binding sites within the MAOA promoter region through JASPAR analysis (relative profile score threshold: 80%), which were validated by ChIP-qPCR (Fig. EV4G). This analysis confirmed specific binding of P65 to Site 2 of the MAOA promoter (Fig. EV4H,I). These findings suggested that the HER2 S310F mutation led to reduced P65 nuclear translocation and binding to MAOA promoter, and inhibited MAOA transcription.

## Palonosetron inhibits Ca²⁺/CAMKK2/AMPK signaling to promote ferroptosis

HER2 S310F mutant cells exhibited an accumulation of 5-HT, which led to aumolertinib resistance. Therefore, the mechanisms by which 5-HT mediates resistance to aumolertinib warrant further investigation. Gene Set Enrichment Analysis (GSEA) of transcriptomic sequencing data indicated that the cellular response to oxidative stress was attenuated in HER2 S310F mutant cells (Fig. 5A). Transient variations in ROS play crucial regulatory roles; however, when ROS production surpasses the cellular antioxidant capacity, then oxidative stress occurs (Guo et al, 2023). Ferroptosis is a regulated form of cell death that is dependent on iron and ROS,

and ferroptosis-inducing therapy shows promise in overcoming resistance to EGFR-TKI (Zhang et al, 2021). Therefore, we assessed the levels of lipid ROS and ferroptosis biomarkers, GPX4 and xCT, in EV and HER2 S310F mutant cells treated with aumolertinib. The findings revealed a reduction in lipid ROS levels in HER2 S310F mutant cells following aumolertinib treatment (Fig. 5B). By contrast, GPX4 and xCT levels were significantly diminished in EV cells following aumolertinib treatment, and no substantial changes were observed in HER2 S310F mutant cells (Fig. 5C,D). These results suggested that HER2 S310F mutant cells showed resistance to ferroptosis when treated solely with aumolertinib. Moreover, the combination of palonosetron and aumolertinib in HER2 S310F mutant cells resulted in increased lipid ROS levels alongside decreased expression of GPX4 and xCT (Fig. 5E,F), suggesting that this drug combination promoted ferroptosis.

To further elucidate whether the observed resistance to ferroptosis in HER2 S310F mutant cells during aumolertinib treatment is related to elevated 5-HT levels, we treated HER2 S310F mutant cells with ferroptosis inducer RSL3 with and without exogenous 5-HT supplementation. The results demonstrated that exogenous 5-HT significantly increased $IC_{50}$ of RSL3 and counteracted the RSL3-induced rise in lipid ROS levels (Fig. 5G,H). These findings suggested that 5-HT could suppress the initiation of ferroptosis. 5-HT can promote Ca²⁺ influx by activating HTR3, and HTR3 antagonists can effectively inhibit 5-HT-induced Ca²⁺ influx (Renden et al, 2024). Furthermore, Ca²⁺ is closely associated with

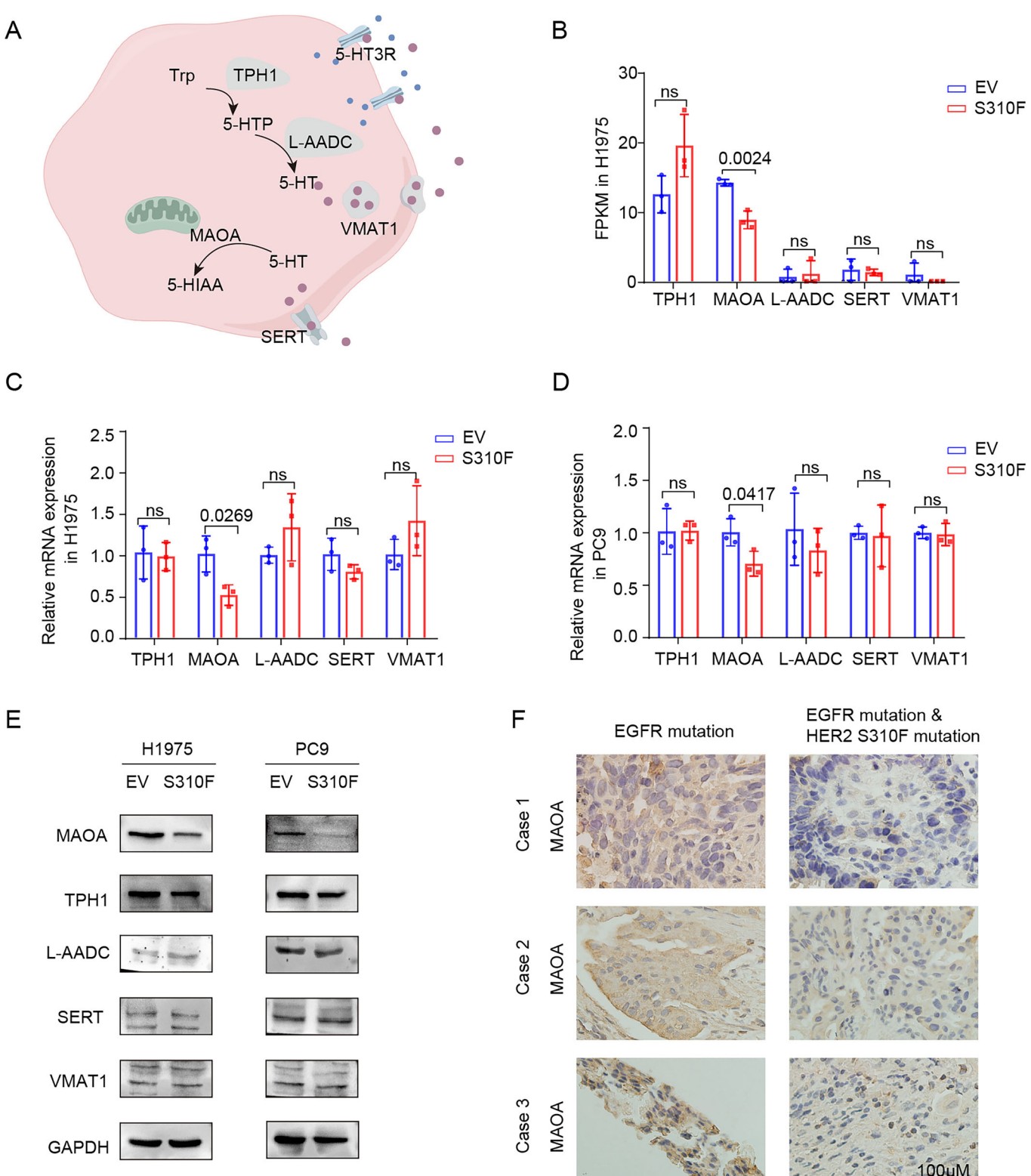

**Figure 4. HER2 S310F mutation induces the reduced expression of MAOA.**

(A) 5-HT synthesis, transport, and metabolic pattern diagram was illustrated by Figdraw. (B) RNA seq analysis of 5-HT metabolism-related gene expression ($n = 3$). (C, D) qPCR analysis of 5-HT metabolism-related gene expression in H1975 EV/S310F and PC9 EV/S310F cells ($n = 3$). (E) Relative protein levels of 5-HT metabolism modulators. (F) MAOA levels in tissues of patients with EGFR mutations or EGFR and HER2 S310F co-mutations. Magnification, ×400; scale bar, 100 μm. Data are presented as mean ± SD. Statistical test: two-tailed unpaired Student's $t$ test. Source data are available online for this figure.

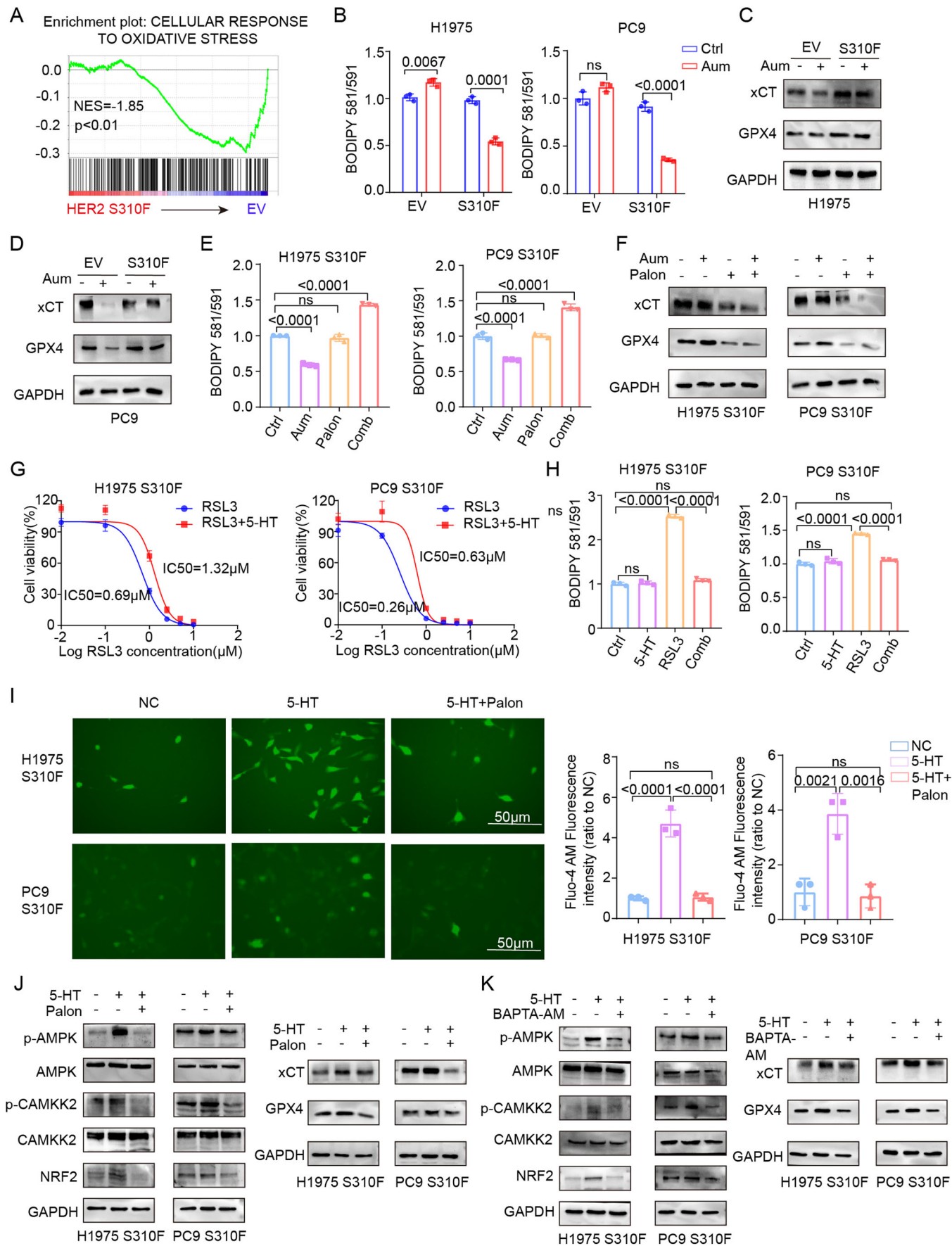

**Figure 5.   5-HT targeting Ca²⁺/CAMKK2/AMPK pathway mediates aumolertinib resistance through inducing resistance to ferroptosis.**

(**A**) Perform GSEA analysis on H1975 EV/S310F RNA seq data. (**B**) The lipid ROS levels after treatment with aumolertinib in H1975 EV/S310F and PC9 EV/S310F cells for 72 h ($n = 3$). (**C, D**) The protein levels of ferroptosis biomarkers after treatment with aumolertinib for 72 h. (**E**) The lipid ROS levels after treatment with aumolertinib, palonosetron alone or in combination in H1975 S310F and PC9 S310F cells for 72 h ($n = 3$). (**F**) The protein levels of ferroptosis biomarkers after treatment with aumolertinib, palonosetron alone, or in combination for 72 h. (**G**) The effect of 5-HT on the $IC_{50}$ of RSL3 ($n = 4$). (**H**) The lipid ROS levels after treatment with 5-HT, RSL3 alone, or in combination for 72 h ($n = 3$). (**I**) Representative images of Fluo-4 AM fluorescence ($n = 3$). Scale bar, 50 μm. (**J**) The protein levels of phosphorylated and total CAMKK2 and AMPK, NRF2 proteins, and ferroptosis biomarkers after treatment with 5-HT alone or a combination of 5-HT and palonosetron for 72 h. (**K**) The protein levels of phosphorylated and total CAMKK2 and AMPK, NRF2 proteins, and ferroptosis biomarkers after treatment with 5-HT alone or a combination of 5-HT and palonosetron for 72 h. Data are presented as mean ± SD. Two-tailed unpaired Student's $t$ test was used for statistical analysis in (**B**), one-way ANOVA was used for statistical analysis in (**E, H, I**). Source data are available online for this figure.

ferroptosis (Wu et al, 2024). We verified that the intracellular Ca²⁺ concentration increased after 5-HT treatment, which was counteracted by the addition of palonosetron (Fig. 5I). Previous studies have demonstrated that CAMKK2 modulates ferroptosis through the AMPK/NRF2 axis (Wang et al, 2022). We further proved that the Ca²⁺ downstream signaling pathway, activated by 5-HT treatment, increased the levels of CAMKK2, AMPK phosphorylation, and NRF2 expression, thereby increasing resistance to ferroptosis. Conversely, palonosetron counteracted the effects of 5-HT to promote ferroptosis (Fig. 5J). Further validation via NRF2 inhibitor ML385 demonstrates that ML385 administration reduced NRF2, xCT, and GPX4 expression while elevating lipid ROS levels, collectively supporting the critical involvement of NRF2 in 5-HT-mediated ferroptosis resistance (Appendix Fig. S4). To determine if the changes in downstream signaling pathways were due to Ca²⁺ concentrations, we employed the Ca²⁺ chelator BAPTA-AM. BAPTA-AM treatment counteracted the increase in 5-HT-induced phosphorylation levels of CAMKK2 and AMPK, as well as changes in NRF, xCT, and GPX4 (Fig. 5K). Furthermore, we experimentally excluded the involvement of autophagy-dependent ferroptosis regulation by 5-HT and palonosetron (Appendix Fig. S5). These findings suggested that elevated levels of 5-HT in HER2 S310F mutant cells, coupled with Ca²⁺ influx, contributed to ferroptosis resistance. Palonosetron treatment inhibited 5-HT binding to the 5-HT3 receptor, obstructed Ca²⁺ influx, suppressed downstream signaling pathways that promote ferroptosis, and reversed aumolertinib resistance.

## Palonosetron enhances the antitumor effect of aumolertinib in HER2 S310F mutant LUAD in vivo

On the basis of the reversal of aumolertinib resistance by palonosetron in vitro, we estimated whether similar therapeutic effects may occur in a subcutaneous xenograft model. PC9 S310F mutant cells were subcutaneously injected into the left inguinal region of BALB/c nude mice. When the tumor size reached an average of 150–200 mm³, the mice were randomly divided into four groups: control, aumolertinib, palonosetron and aumolertinib + palonosetron (Fig. 6A). The primary tumors are shown in Fig. 6B. In accordance with the results in vitro, the combination treatment exhibited the greatest inhibitory effects on tumor volume and weight compared with the three other treatment groups, without serious toxicity in terms of body weight change (Fig. 6C–E). In addition, we found no significant increase in blood urea nitrogen/ creatinine and aspartate aminotransferase/alanine aminotransferase levels, suggesting normal kidney and liver functions (Fig. 6F–I). Hematoxylin and eosin (H&E) staining was performed on cardiac, hepatic, and renal tissues from control mice and those administered

aumolertinib monotherapy, palonosetron monotherapy, or combination therapy. Histopathological analysis revealed no evident morphological alterations in any treatment cohort compared to controls (Appendix Fig. S6). These findings corroborate the absence of overt toxicological effects associated with the combination of aumolertinib and palonosetron. The results of IHC showed that the lowest level of xCT and p-CAMKK2 expression was found in the two-drug combination treatment group (Fig. 6J). These results indicated that the combination of aumolertinib and palonosetron not only maintained optimal antitumor efficacy by promoting ferroptosis but also ensured safety and minimal toxicity in vivo.

## 5-HT mediates secondary resistance to aumolertinib

To further investigate whether palonosetron exerts a similar effect in aumolertinib secondary resistance cells, we established H1975 aumolertinib secondary resistance cells, H1975 AR. The results from CCK8 assays indicated a significant increase in $IC_{50}$ of aumolertinib in H1975 AR cells (from 0.39 to 5.85 μM), confirming that secondary resistance cells were constructed successfully (Fig. 7A). Treatment of H1975 AR cells with aumolertinib, palonosetron, or a combination of the two drugs revealed a notable synergistic effect across most concentration combinations, as indicated by the CI (Fig. 7B). In addition, the application of fixed concentrations of palonosetron markedly decreased $IC_{50}$ of aumolertinib in H1975 AR cells (Fig. 7C). We further evaluated the effect of the two drugs on colony formation. The combination of palonosetron and aumolertinib demonstrated significantly stronger inhibition of colony formation than either drug alone (Fig. 7D).

We verified whether 5-HT aggregation is also present in H1975 AR cells. Notably, we found no alteration in HTR3 expression in H1975 AR cells compared with wild-type cells, but the concentration of 5-HT in the culture medium was elevated (Fig. 7E,F). We collected blood samples from seven patients before EGFR-TKI treatment (Pre-treatment) and six patients following acquired EGFR-TKI resistance (Post-treatment). Analysis demonstrated significantly elevated 5-HT levels in resistant patients (Fig. 7G). Furthermore, we confirmed the downregulation of MAOA expression in H1975 AR cells through western blot analysis (Fig. 7H). We also collected computed tomography images of two patients treated with third-generation EGFR-TKI in three stages: the tumor prior to EGFR-TKI administration, the tumor reduction during EGFR-TKI therapy, and the subsequent tumor progression following the development of EGFR-TKI resistance (Fig. 7I). We obtained tissue specimens from these two patients before and after third-generation EGFR-TKI treatment, and MAOA expression was detected using IHC, which showed a significant reduction in MAOA expression in the tissues after drug resistance (Fig. 7J). These results suggested that following

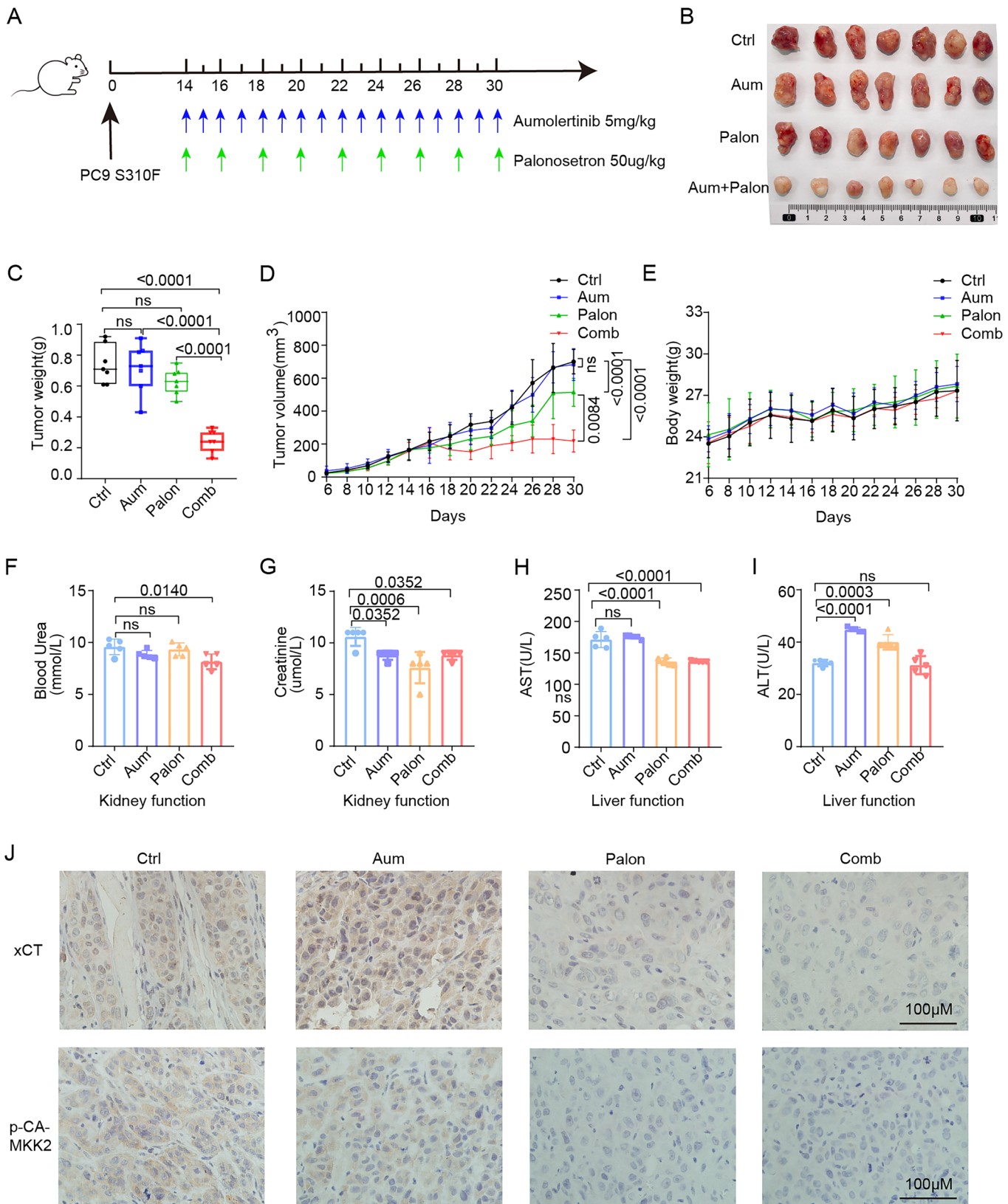

**Figure 6. Combination of aumolertinib and palonosetron significantly inhibits tumor growth in vivo.**

(A) Diagram of mouse model treated with aumolertinib and palonosetron. (B) Representative images of tumors after inoculation using PC9 S310F cells treated with aumolertinib, palonosetron, or the combination. (C) The tumor weights in the four groups after treatment ($n = 7$). The boxes extend from the 25th to 75th percentile, the middle line shows the median, whiskers extend to the most extreme data. (D) Time course of tumor volume ($n = 7$). (E) Time course of mice weight ($n = 7$). (F–I) The kidney and liver function in mice bearing PC9 S310F xenografts after treatment ($n = 5$). (J) IHC analysis of xCT, NRF2, and p-CAMKK2 protein expression was performed using tumor sections of PC9 S310F mouse xenografts treated as indicated above. Magnification, ×400; scale bar, 100 μm. Data are presented as mean ± SD. Statistical test: one-way ANOVA. Source data are available online for this figure.

secondary resistance to EGFR-TKI, 5-HT levels increased and MAOA expression decreased. Organoids retain both histopathological features of parental tumors and their sensitivity to targeted therapies, thereby enabling validation and discovery of biomarker-therapeutic associations (Shi et al, 2020). We collected postoperative tissues from 3 patients with lung adenocarcinoma and performed organoid culture. Following 10 days of culture, organoids were divided into four groups: vehicle control, aumolertinib monotherapy, aumolertinib + 5-HT, and aumolertinib + 5-HT + palonosetron. After 72 h of drug treatment, the CCK8 test results showed that 5-HT significantly attenuated the cytotoxic efficacy of aumolertinib in patient-derived lung cancer organoids, while the addition of palonosetron treatment counteracted the effect of 5-HT (Appendix Fig. S7). Thus, plasma 5-HT levels may serve as a potential biomarker for third-generation EGFR-TKI resistance, and the treatment of HTR3 antagonist palonosetron can reverse secondary resistance to aumolertinib.

# Discussion

Compared with first-generation EGFR-TKI as first-line therapy, third-generation EGFR-TKI can improve median PFS (Soria et al, 2018; Lu et al, 2022). In addition, compared with standard chemotherapy in patients with T790M-driven acquired resistance to first-generation EGFR-TKI, third-generation EGFR-TKI exhibit a superior objective response rate and PFS (Mok et al, 2017). These results reflect the importance of third-generation EGFR-TKI in the treatment of patients with EGFR mutant LUAD. The treatment strategy after third-generation EGFR-TKI resistance is based on traditional chemotherapy, but the efficacy of chemotherapy is limited and far from meeting the clinical needs (Schmid et al, 2020; Tanaka et al, 2021). Therefore, new biomarkers and individualized treatment regimens based on different resistance mechanisms must be explored to prolong the survival of patients and improve their quality of life.

HER2 S310F, an oncogenic mutation, promotes HER2 receptor dimerization and activation of downstream signaling pathways (Ishiyama et al, 2023). Patients with HER2 S310F mutation in biliary tract cancer and upper urinary tract epithelial cancer have poor sensitivity to HER2-TKI (Kim et al, 2020; Harding et al, 2023). In colorectal cancer, HER2 S310F mutation is also ineffective against dual HER2/EGFR-targeting TKI (Vaghi et al, 2023). However, the effect of the HER2 S310F mutation on third-generation EGFR-TKI sensitivity in LUAD is unclear. Here, we collected clinical cases with HER2 S310F mutation and validated through in vitro and in vivo experiments that HER2 S310F mutation mediated third-generation EGFR-TKI resistance in LUAD. Further research has revealed that HER2 S310F mutant cells exhibit increased accumulation of the neurotransmitter 5-HT, which subsequently mediates resistance to aumolertinib.

5-HT serves crucial physiological roles in the central nervous system and the peripheral multi-system. Various 5-HT receptors have been shown to be expressed in the lungs (Nikolić et al, 2023). 5-HT is involved in the occurrence and development of various lung diseases, including asthma, pulmonary fibrosis, fungal pneumonia, and lung cancer (Konigshoff et al, 2010; Mendez-Enriquez et al, 2021; Renga et al, 2023; Legchenko et al, 2024; Zheng et al, 2024). Although many cancers utilize 5-HT as an autocrine or paracrine growth factor, its role in modulating drug resistance is less well understood (Chen et al, 2024). In our study, we observed that elevated levels of 5-HT were associated with the development of resistance to aumolertinib. Further investigation of the cause of elevated 5-HT levels revealed that its metabolizing enzyme, MAOA, was significantly reduced in drug-resistant cells. Reduced MAOA expression correlates with aggressive malignant phenotypes and poor clinical prognosis across multiple cancers, including gastric, hepatocellular, and cholangiocarcinoma (Alpini et al, 2008; Pang et al, 2020; Wang et al, 2023). Conversely, elevated MAOA expression in prostate cancer promotes tumor growth, metastasis, stemness, and therapy resistance (Li et al, 2020). Emerging evidence further implicates MAOA in modulating the tumor immune microenvironment, underscoring its multifaceted role in tumors (Wang et al, 2021; Zhao et al, 2025). Nevertheless, the relationship between MAOA and therapeutic resistance remains underexplored, with only one study reporting crosstalk between MAOA and androgen receptor expression in prostate cancer. Genetic or pharmacologic targeting of MAOA enhanced the growth-inhibition efficacy of enzalutamide, darolutamide, and apalutamide in both androgen-dependent and castration-resistant prostate cancer cells (Wei et al, 2021). In our study, we confirmed the important role of MAOA in EGFR-TKI resistance by knocking down MAOA in H1975, PC9 wild-type cells and overexpressing MAOA in H1975 S310F and PC9 S310F cells. In addition, we collected tissue samples from three patients with HER2 S310F and EGFR co-mutations, as well as two pairs of tissue samples from patients before and after third-generation EGFR-TKI treatment. IHC of MAOA expression showed a significant decrease in co-mutated and secondary resistant tissue samples. These results suggested that MAOA may still be a potential biomarker for third-generation EGFR-TKI resistance. However, in the future, additional patient tissue samples need to be collected for validation.

Drug repositioning is a strategy that utilizes an existing therapeutic to a new disease indication, and it holds the promise of rapid clinical impact at a lower cost than de novo drug development (Ashburn and Thor, 2004). Growing evidence supports targeting HTRs as a novel anticancer strategy, with pharmacological inhibition of HTR2B suppressing tumor proliferation across multiple malignancies, including gastric, colorectal, pancreatic, and hepatocellular carcinomas (Soll et al, 2010; Jiang et al, 2017; Liu et al, 2023; Tu et al, 2023). Furthermore,

pharmacological targeting of HTR3 inhibits proliferation in lung adenocarcinoma (Tone et al, 2020). In our study, we screened and identified HTR3 antagonists that reverse third-generation EGFR-TKI resistance. The introduction of HTR3 antagonists into clinical use revolutionized the treatment of nausea and vomiting in cancer patients receiving chemo- or radiation therapy. Moreover, they are generally well tolerated by patients, with only a restricted range of side effects (Machu, 2011). There is preclinical evidence supporting the antitumor effects of HTR3 antagonists in colorectal cancer and lung cancer (Rashidi et al, 2019; Li et al, 2021). Further studies should test the combination of HTR3 antagonists and third-generation EGFR-TKI in trials, especially in patients with third-generation EGFR-TKI-resistant LUAD. The combination therapy strategy proposed here may have broad implications for overcoming third-generation EGFR-TKI resistance by enabling personalized treatment regimens based on patient-specific 5-HT levels as a predictive biomarker.

A previous study showed that 5-HT mediates ferroptosis resistance (Liu et al, 2023). We revealed that aumolertinib-resistant cells exhibited ferroptosis resistance due to increased accumulation of 5-HT. Consistent with previous reports (Li et al, 2021), the binding of 5-HT to HTR3 resulted in the opening of ion channels, influx of $Ca^{2+}$, which could be attributed to the activation of the $Ca^{2+}$/CAMKK2/AMPK pathway. Accumulating evidence has revealed that the $Ca^{2+}$/CAMKK2/AMPK pathway is closely related to oxidative stress (Wang et al, 2022; Yang et al, 2024). Here, we proved that activation of the CAMKK2/AMPK pathway by $Ca^{2+}$ influx further mediated resistance to ferroptosis, and treatment with HTR3 antagonists or $Ca^{2+}$ chelating agents could counteract a series of reactions caused by increased 5-HT. However, the specific regulatory mechanism still needs in-depth research in the future.

In summary, our study suggested that the accumulation of 5-HT promotes resistance to ferroptosis, partially mediated by the $Ca^{2+}$/CAMKK2/AMPK signaling pathway, which contributes to aumolertinib resistance. The combination of aumolertinib with HTR3 antagonists effectively overcame aumolertinib resistance. This discovery not only enriches our comprehension of the mechanisms underlying EGFR-TKI resistance but also has the potential to provide an interdisciplinary theoretical basis for the development of new treatment strategies.

# Methods

### Reagents and tools table

| Reagent/resource | Reference or source | Identifier or catalog number |
| --- | --- | --- |
| **Experimental models** | | |
| Balb/c nude | GemPharmatech | |
| H1975 cells | American Type Culture Collection | CRL-5908 |
| HEK293T cells | American Type Culture Collection | CRL-11268 |
| PC9 cells | Immcell | IM-H125 |
| H3255 cells | Immcell | IM-H122 |
| Patients' samples | Tianjin Medical University Cancer Institute and Hospital | |
| **Recombinant DNA** | | |

| Reagent/resource | Reference or source | Identifier or catalog number |
| --- | --- | --- |
| PCDH-vector-CMV-MCS-EF1-puro | Suzhou Hong Xun Biotechnology | |
| PCDH-HER2 (human)-CMV-MCS-EF1-puro | Suzhou Hong Xun Biotechnology | |
| Plsi-ctrl-vector | Suzhou Hong Xun Biotechnology | |
| pCMV-MAOA (human)-3×FLAG-Neo | Wuhan miaolingbio Bioscience & Technology | P49954 |
| **Antibodies** | | |
| Rabbit anti-HER2 | Cell Signaling Technology | 2165S |
| Rabbit anti-HTR3A | Proteintech | 29685-1-AP |
| Rabbit anti-MAOA | Proteintech | 10539-1-AP |
| Rabbit anti-TPH1 | Abcam | ab52954 |
| Rabbit anti-AADC | Proteintech | 10166-1-AP |
| Rabbit anti-SERT | Proteintech | 19559-1-AP |
| Rabbit anti-VMAT1 | Proteintech | 20340-1-AP |
| Rabbit anti-P65 | Proteintech | 10745-1-AP |
| Rabbit anti-LaminB1 | Proteintech | 12987-1-AP |
| Rabbit anti-xCT | Cell Signaling Technology | 12691S |
| Rabbit anti-GPX4 | Cell Signaling Technology | 52455S |
| Rabbit anti-AMPK | Proteintech | 10929-2-AP |
| Rabbit anti-p-AMPK | Affinity Biosciences | AF3423 |
| Rabbit anti-CAMKK2 | Proteintech | 11549-1-AP |
| Rabbit anti-p-CAMKK2 | Affinity Biosciences | AF4487 |
| Rabbit anti-NRF2 | ABclonal | A0674 |
| Rabbit anti-LC3B | Cell Signaling Technology | 3868S |
| Rabbit anti-ATG5 | Cell Signaling Technology | 12994S |
| Rabbit anti-P62 | Cell Signaling Technology | 23214S |
| Rabbit anti-Beclin1 | Cell Signaling Technology | 3495S |
| Mouse anti-GAPDH | Santa Cruz Biotechnology | sc-47724 |
| mouse anti-rabbit IgG-HRP | Santa Cruz Biotechnology | sc-2357 |
| m-IgGκ BP-HRP | Santa Cruz Biotechnology | sc-516102 |
| Alexa Fluor™ 647 | Invitrogen | P21462 |
| **Oligonucleotides and other sequence-based reagents** | | |
| Sequences of lentivirus or plasmids | Beijing Liuhe BGI Co., Ltd | Table EV4 |
| qRT-PCR primers | Beijing Liuhe BGI Co., Ltd | Table EV5 |
| ChIP-qPCR primers | Beijing Liuhe BGI Co., Ltd | Table EV6 |
| **Chemicals, enzymes and other reagents** | | |
| PRMI 1640 | Gibco | 11875500 |
| DMEM | Gibco | 11995500 |
| FBS | PAN Seratech | ST30-3302p |
| Penicillin-streptomycin | Gibco | 15140-122 |
| Puromycin | Gibco | A1113803 |
| Hygromycin B | TargetMol | T1022 |

| Reagent/resource | Reference or source | Identifier or catalog number |
|---|---|---|
| Aumolertinib | Hansoh Pharmaceutical Group Company Limited in China | |
| Osimertinib | Selleck | S7297 |
| Palonosetron HCl | Selleck | S3050 |
| FDA-approved Drug Library | Selleck | L1300 |
| Serotonin (5-HT) HCl | Selleck | S4244 |
| RSL3 | Selleck | S8155 |
| BAPTA-AM | MCE | HY-100545 |
| BODIPY™ 581/591 C11 | Invitrogen | D3861 |
| ML385 | Selleck | S8790 |
| Cell Counting Kit-8 | bioshrp | BS350B |
| 5-HT(Serotonin/5-Hydroxytryptamine) ELISA Kit | Elabscience | E-EL-0033 |
| NE-PER™ Nuclear and Cytoplasmic Extraction Reagents | Thermo Scientific | 78835 |
| BeyoChIP ChIP Assay Kit | Beyotime Biotechnology | P2080S |
| PCR Clean Up Kit | Beyotime Biotechnology | D0033 |
| Fluo-4 AM | Beyotime Biotechnology | S1060 |
| Albumin Bovine V | GeneRun | A0211 |
| Immobilon Western HRP Substrate | Millipore | WBKLS0500 |
| SPARKeasy Cell RNA Rapid Extraction Kit | SparkJade | AC0205 |
| ChamQ SYBR qPCR Master Mix | Vazyme | Q341 |
| NucBlue™ Fixed Cell ReadyProbes™(DAPI) | Invitrogen | R37606 |
| **Software** | | |
| GraphPad Prism 8 | GraphPad Software | |
| CytExpert | | |
| CompuSyn | | |
| Adobe Illustrator | Adobe Systems | |
| JASPAR | | |
| Figdraw | | |
| **Other** | | |

## Cell culture and reagents

The H1975 (ATCC, CRL-5908) and HEK293T (ATCC, CRL-11268) cell lines were purchased from ATCC. The PC9 (Immcell, IM-H125) and H3255 (Immcell, IM-H122) cell lines were purchased from Immcell. All human LUAD cell lines were cultured in RPMI 1640 medium (Gibco, Cat# 11875500) with 10% fetal bovine serum (PAN Seratech, Cat# ST30-3302p) and 1% penicillin/streptomycin (Gibco, Cat# 15140-122). The HEK 293T cells were cultured in DMEM (Gibco, Cat# 11995500) with 10% fetal bovine serum and 1% penicillin/streptomycin. All cell lines were incubated at 37 °C in a humidified

incubator with 5% $CO_2$. Cell lines were not authenticated as they were directly procured from vendors. All cells were mycoplasma-free.

## Plasmid construction and cell transfection

Full-length human HA-tagged HER2 was acquired from Suzhou Hong Xun Biotechnology Co., Ltd. in China. The primer sequences for the HER2 S310F point mutation and targeting sequences for HTR3A and MAOA knockdown are provided in Appendix Table S3. All constructs were validated through sequencing. The packaging and expression plasmids were transfected into HEK293T cells. After 48 h, the supernatant was collected. Target cells were subsequently infected with lentivirus expressing particles and selected with puromycin for 7–10 days, after which stable cell lines were confirmed.

## High-throughput screening

HTS was performed using the Explorer G3 automated drug screening system from Tianjin Medical University. H1975 empty vector (EV) and H1975 S310F cells were seeded in 384-well plates at a density of 650 cells per well. On the following day, the cells were treated with a library of 2972 FDA-approved compounds (Selleck Chemicals) at a concentration of 10 μM. After 72 h of drug treatment, cell viability was assessed using the Cell Counting Kit-8 (CCK-8, Biosharp, Cat# BS350A) assay. Compounds that resulted in a cell viability of less than 30% compared to the DMSO control were considered sensitive drugs. Candidate compounds were defined as those that exhibited sensitivity in H1975 S310F cells but not in H1975 EV cells.

## Cell viability assays

Cells were plated in 96-well plates and treated with the indicated treatments at varying doses for 48–120 h. After that, 100 μL of fresh medium containing 10 μL of CCK-8 solution was added to each well, followed by further incubation in the dark for 2–3 h at 37 °C. The absorbance of the wavelength at 450 nm was measured. The percentage of cell viability was calculated as follows: percentage of cell viability = $(OD_{treatment} - OD_{blank})/(OD_{control} - OD_{blank})$ 100%. 100%. Furthermore, the combination index (CI) was calculated using CompuSyn Software (CompuSyn, Inc., Paramus, NJ, USA).

## Colony formation assay

About 1000 cells were plated into a 12-well plate and cultured in complete medium for the colony formation assay. After 10 days of incubation, the colonies were rinsed, fixed, and stained with 0.5% crystal violet, imaged, and counted.

## Establishing aumolertinib acquired resistance cell lines

H1975 aumolertinib secondary resistance cells were generated from parental H1975 cells via stepwise dose escalation over 24 weeks: beginning at 10 nM in week 2 and progressively increasing to a final concentration of 10 μM aumolertinib, as previously reported (Yue et al, 2025).

## RNA sequencing (RNA-seq) and analysis

Transcriptome sequencing of H1975 EV/S310F was conducted by Novogene (Beijing, China). DEGs were filtered by log2 (fold

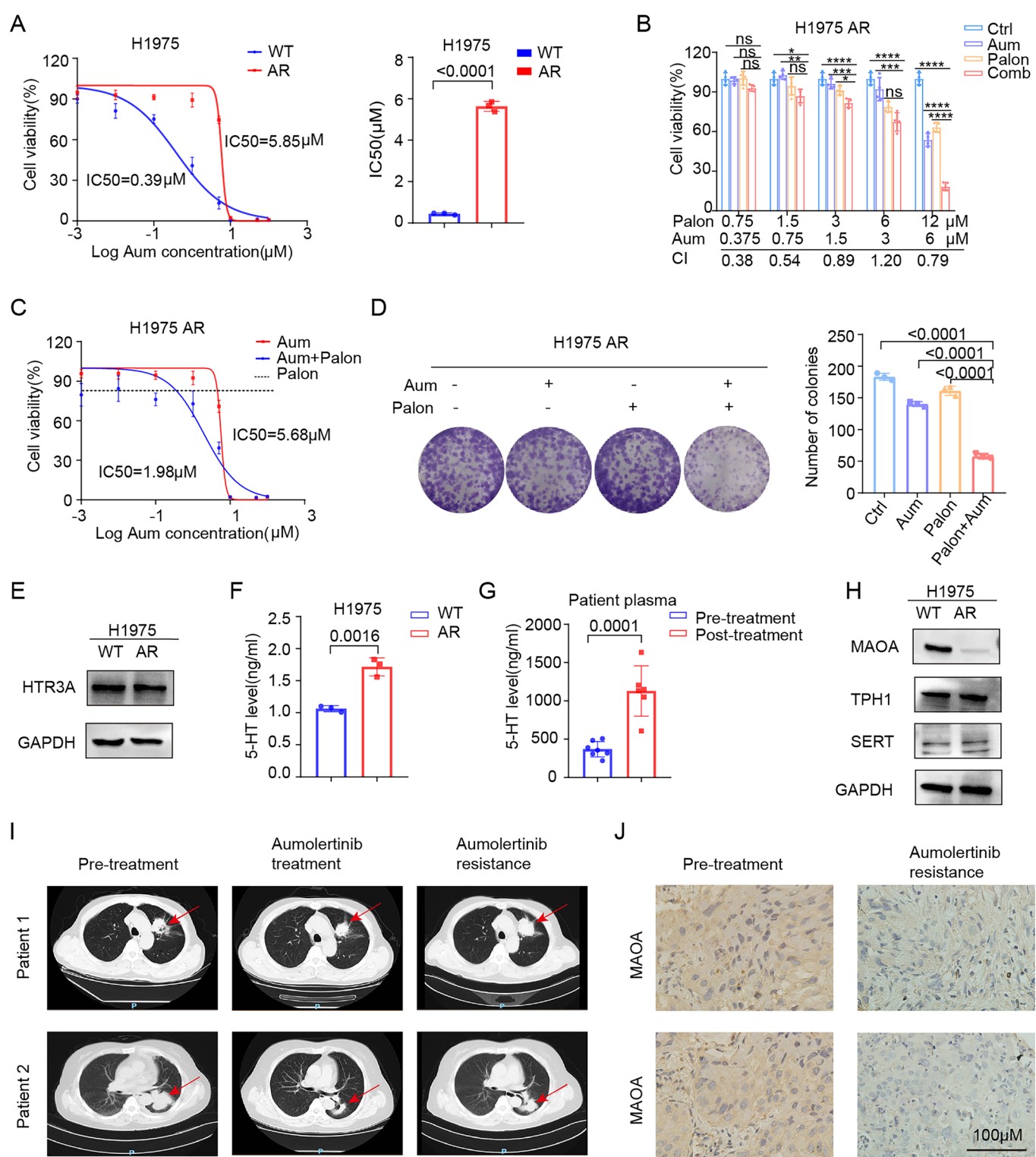

change) ≥1 and P ≤ 0.05. ClusterProfiler was used for gene enrichment analysis of gene ontology and Kyoto Encyclopedia of Genes and Genomes (KEGG), and P ≤ 0.05 was used as the enrichment cut-off.

## Enzyme-linked immunosorbent assay (ELISA)

After the designated treatment, 5-HT and 5-HIAA in the cell supernatant or tumor tissues were detected using commercial kits

◀

**Figure 7. Palonosetron reverses secondary resistance to aumolertinib.**

(A) IC$_{50}$ analysis of aumolertinib by a CCK-8 assay in H1975 EV/S310F cells. Data in the right panel represents mean ± SD of three independent biological replicates. (B) Cell viability and CI analysis in H1975 AR cells treated with indicated concentrations of aumolertinib and palonosetron alone or in combination for 72 h ($n = 4$). P value (Aum 0.75 μM, Palon 1.5 μM): Ctrl vs Comb, $P = 0.0153$, Aum vs Comb, $P = 0.004$; P value (Aum 1.5 μM, Palon 3 μM): Ctrl vs Comb, $P < 0.0001$, Aum vs Comb, $P = 0.0007$, Palon vs Comb, $P = 0.0173$; P value (Aum 3 μM, Palon 6 μM): Ctrl vs Comb, $P < 0.0001$, Aum vs Comb, $P = 0.0007$; P value (Aum 6 μM, Palon 12 μM): Ctrl vs Comb, $P < 0.0001$, Aum vs Comb, $P < 0.0001$, Palon vs Comb, $P < 0.0001$. (C) Dose–response curves determined by the CCK-8 assay were used to calculate the IC$_{50}$ values of aumolertinib for 5 days in the presence or absence of 5 μM palonosetron in H1975 AR cells. Black dotted lines indicate the cell viability at a concentration of 5 μM palonosetron for 5 days ($n = 4$). (D) The colony-forming efficiency of H1975 AR was determined. These cells were treated with aumolertinib or palonosetron, or a combination of both at the same time for 10 days ($n = 3$). (E) HTR3 protein levels in H1975 AR cells. (F) 5-HT levels in the supernatant of H1975 WT/AR cells ($n = 3$). (G) Pre-treatment plasma samples and EGFR-TKI resistance plasma samples (Post-treatment) were collected from LUAD patients for 5-HT detection (Pre-treatment group, $n = 7$; Post-treatment group, $n = 6$). (H) Relative protein levels of 5-HT metabolism modulators in H1975 AR cells. (I) CT images of lung tumor images before and after treatments. The red arrow indicated tumors. (J) MAOA staining in tumor specimens from the same patient before and after treatments. Magnification, ×400; scale bar, 100 μm. Data are presented as mean ± SD. Two-tailed unpaired Student's $t$ test was used for statistical analysis in (A, F, G), one-way ANOVA was used for statistical analysis in (B, D), ns not significant, *$P < 0.05$, **$P < 0.01$, ***$P < 0.001$, and ****$P < 0.0001$.

(Elabscience, Cat# E-EL-0033) based on the recommended protocol in the manufacturer's instructions. Measurements were conducted by using a microplate reader (BioTek, Winooski, VT, USA) at 450 nm.

## Western blot and antibodies

Total protein was isolated with SDS lysis buffer replenished with protease inhibitor cocktail, heated at 95 °C for 10 min, and quantified using a NanoDrop2000 spectrophotometer. The protein samples were separated on 8–12% SDS-PAGE gels and transferred onto a polyvinylidene difluoride membrane, which was accessible to the antibody. Following a blocking step with 5% BSA for 1 h, membranes were cut according to molecular weight markers prior to antibody incubation. When detecting multiple targets from the same sample in western blot, we performed parallel electrophoresis and transfer under identical conditions when physical membrane cutting became impractical. Crucially, GAPDH loading controls were verified for every individual lane. Then, the membranes were incubated with primary antibodies at 4 °C overnight and corresponding secondary antibodies at a 1:4000 dilution for 1 h at room temperature. Primary antibodies against the following targets were used for western blot analyses: anti-HER2 (1:1000, CST, 2165S), anti-HTR3 (1:4000, Proteintech, 29685-1-AP), anti-TPH1 (1:1000, Abcam, ab52954), anti-MAOA (1:1000, Proteintech, 10539-1-AP), anti-SERT (1:500, Proteintech,19559-1-AP), anti-VMAT1 (1:500, Proteintech, 20340-1-AP), anti-AADC (1:500, Proteintech, 10166-1-AP), anti-xCT (1:1000, Proteintech, 26864-1-AP), anti-GPX4 (1:1000, CST, 524555S), anti-NRF2(1:500, ABclonal, A0674), anti-NF-κB p65 (1:1000, Proteintech, 10745-1-AP), anti-CAMKK2 (1:1000, Proteintech, 11549-1-AP), anti-AMPK (1:1000, Proteintech, 10929-2-AP), anti-p-CAMKK2 (Ser511; 1:500, Affinity Biosciences, AF4487), and anti-p-AMPK (Thr172; 1:500, Affinity Biosciences, AF3423).

## Quantitative real-time PCR (qRT-PCR)

Total RNA was isolated using an RNA extraction kit (SparkJade, Cat# AC0205) according to the manufacturer's instructions. Following reverse transcription, real-time PCR assays were performed with SYBR green (Vazyme, Cat# Q341). The primer sequences are listed in Appendix Table S4.

## Immunofluorescence staining

Cells were seeded in 12-well plates containing coverslips and incubated overnight. The next day, cells were fixed with 4%

paraformaldehyde for 15 min, and then followed by 0.1% Triton X-100 permeabilization for 10 min. After blocking with 3% BSA at room temperature for 1 h, and then these cells were incubated at 4 °C overnight with a primary antibodies: anti-NF-κB p65 (1:100, Proteintech, 10745-1-AP). Cells were washed three times with PBS and incubated with fluorescently labeled secondary antibodies for 1 h at room temperature. Then, the nuclei were labeled by DAPI (Invitrogen, Cat# R37606). Finally, cells were visualized using a fluorescence microscope.

## Chromatin immunoprecipitation (ChIP)-qPCR

The JASPAR database (http://jaspar.genereg.net/) was used to predict the potential P65 binding site on the MAOA promoter. ChIP assays using the ChIP assay kit (Beyotime, Cat# P2080S) validated the binding regions of P65 on the MAOA promoter according to the manufacturer's instructions. Briefly, the cells were cross-linked, lysed, and sonicated into appropriate fragments. The prepared chromatin was precipitated using IgG or ChIP-grade antibody against P65 at 4 °C overnight and then incubated with Protein A + G Agarose/Salmon Sperm DNA at 4 °C for 1 h. Then, the binding complexes were thoroughly washed, eluted, purified, and analyzed by qRT-PCR. The primer sequences used for the ChIP-qPCR assay in the MAOA promoter region were provided in Appendix Table S5.

## Ca$^{2+}$ fluorescent probes

Cells were plated in 96-well plates and incubated with the indicated treatments. To detect intracellular Ca$^{2+}$, cells were treated with 1 μM of Fluo-4 AM (Beyotime, Cat# S1060) fluorescent probes and incubated in the dark for 30 min. Subsequently, cells were washed three times with PBS. The intracellular Ca$^{2+}$ levels were quantified by measuring the fluorescence intensity of Fluo-4 AM.

## Lipid ROS analysis

Cells were plated in six-well plates and incubated with the indicated treatments. Thereafter, the culture media were replaced with serum-free media containing 2 μM C11-BODIPY (581/591, Invitrogen, Cat# D3861) for staining, and cells were incubated in the dark for 30 min to determine the lipid ROS levels. Cells were analyzed using a flow cytometer (BD Biosciences) equipped with an excitation wavelength of 488 nm and an emission wavelength of 525 nm.

## Animal models

Male BALB/C nude mice (5 weeks old, 18–20 g) were purchased from Jiangsu GemPharmatech Co., Ltd. (Jiangsu, China). Approximately $1 \times 10^6$ PC9 EV/S310F cells that had been resuspended in 100 μL of PBS were injected subcutaneously into the right flanks of male BALB/c nude mice. The tumor volume and weight of the mice were observed every 2 days. Tumor volume (V) was calculated using the following formula: length × width$^2$ × 1/2. The mice were randomly divided into four groups when the tumor volumes reached ~150–200 mm$^3$. The mice were given a regular chow diet and water and accommodated under specific pathogen-free conditions in a standard laboratory environment (21 ± 2 °C, 12 h light/dark cycle). All experimental procedures in this study were approved by the Animal Ethical and Welfare Committee of Tianjin Medical University Cancer Institute and Hospital.

## Clinical samples

Clinical specimens utilized in this study encompassed three distinct types: freshly resected postoperative tumor tissues, peripheral blood samples, and formalin-fixed paraffin-embedded tissue blocks. Our study included both male and female patients. Key inclusion criteria were: (1) Pathologically confirmed lung adenocarcinoma; (2) Aged ≥18 and ≤80 years. Key exclusion criteria were: (1) Histopathological diagnosis of non-lung adenocarcinoma; (2) Presence of synchronous or metachronous primary malignancies (other than lung adenocarcinoma). All experiments conducted in patients conformed to the principles set out in the WMA Declaration of Helsinki and the Department of Health and Human Services Belmont Report. All experiments were approved by the Ethics Committee of Tianjin Medical University Cancer Hospital (approval number: E20250575). All participants have provided written informed consent to take part in the study.

## Immunohistochemistry (IHC)

Paraffin-embedded LUAD slices were acquired from the Tianjin Medical University Cancer Institute and Hospital. Xenograft specimens derived from the PC9 S310F cell line were fixed in formalin and embedded in paraffin. The primary antibodies used for IHC were anti-MAOA (1:400, Proteintech, 10539-1-AP), anti-p-CAMKK2 (Ser511; 1:100, Affinity Biosciences, AF4487), and anti-xCT (1:100, Proteintech, 26864-1-AP).

## Toxicity study

For toxicity assessment, blood samples of mice were collected from the orbital sinus after treatment and subjected to biochemical analysis for kidney function markers blood urea nitrogen (BUN) and creatinine and liver marker enzymes alanine transaminase (ALT) and aspartate transaminase (AST) by COBAS® 8000 analyzer series (Roche Diagnostics, Rotkreuz, Switzerland). In addition, the hearts, livers, and kidneys of mice were freshly isolated, fixed in 4% paraformaldehyde, paraffin-embedded, and sectioned. The sections underwent sequential dewaxing in xylene, graded ethanol hydration, and hematoxylin and eosin (H&E) staining. Following graded ethanol dehydration and xylene clearance, coverslipped slides were observed by bright-field microscopy. The study was approved by the Research

**The paper explained**

**Problem**

While LUAD patients with activating mutations in the EGFR clearly benefit from EGFR-TKI therapy, substantial interpatient heterogeneity in treatment response persists. It is urgent to explore personalized treatment strategies based on distinct resistance mechanisms to reverse EGFR-TKI resistance.

**Results**

We report that HER2 S310F mutation contributes to third-generation EGFR-TKI resistance, mechanistically linked to aberrant accumulation of 5-HT. Specifically, the accumulation of 5-HT promotes resistance to ferroptosis, partially mediated by the Ca2 + /CAMKK2/AMPK signaling pathway, which contributes to aumolertinib resistance. Importantly, combinatorial administration of aumolertinib with palonosetron, an FDA-approved HTR3 antagonist, effectively reversed drug resistance in preclinical models, demonstrating therapeutic potential.

**Impact**

Our findings support plasma 5-HT levels as a clinically actionable biomarker for predicting EGFR-TKI resistance. The combination of aumolertinib with palonosetron is a promising therapeutic approach for aumolertinib resistance induced by 5-HT accumulation.

Ethics Committee of the Tianjin Medical University Cancer Institute and Hospital (approval no. AE2023023).

## Tissue processing and organoid culture

Lung cancer organoids were derived from surgery samples of lung adenocarcinoma patients at Tianjin Medical University Cancer Institute and Hospital. The study was approved by the Ethical Committee of Tianjin Medical University Cancer Institute and Hospital. All patients participating in this study provided signed informed consent.

Fresh surgical specimens were washed twice with F12 containing triple antibiotics, minced in organoid medium, and non-neoplastic tissue was removed. Tumor fragments were digested in 10 mL Miltenyi digestion buffer at 37 °C for 30–40 min, and microscopic monitored. Upon single-cell suspension confirmation, samples were filtered through a 100-μm mesh into a 50-mL tube, rinsed with F12 to 25 mL. After centrifugation (300 × g, 5 min), erythrocyte-rich pellets were lysed in 5 mL ACK buffer (5 min, ice), re-centrifuged, and pellets resuspended in 1 mL F12 for counting. Following F12 wash (25 mL final volume), cells were resuspended in complete organoid medium (500 μL–1 mL) to $5–7 \times 10^6$ cells/mL. Cell suspension was mixed 1:1 with Matrigel and plated 10 μL/well in 96-well plates. Culture plates were incubated 1 h, inverted before adding 37 °C-preheated organoid medium. Organoids were monitored daily, and culture media were added or changed every 3 days during the cultivation process.

## Statistical analysis

No blinding was implemented in the experimental design. Mice were randomized by stratification based on tumor volume and body weight, with sample sizes estimated according to previous studies. Heatmap was plotted by https://www.bioinformatics.com.cn (last accessed on 10 December 2024), an online platform for data

analysis and visualization (Tang et al, 2023). All data were obtained from at least three independent experiments and expressed as mean ± SD. The software program GraphPad Prism 8 (San Diego, CA, USA) was used to analyze the quantitative results. Statistical significance was determined with two-sided Student's $t$ test or one-way ANOVA test according to different conditions. ns not significant, $*P < 0.05$, $**P < 0.01$, $***P < 0.001$ and $****P < 0.0001$. Kaplan–Meier functions were used to illustrate survival profiles using Mantel–Cox tests.

## Data availability

RNA sequencing data were deposited in the Gene Expression Omnibus platform with accession number GSE296909.

The source data of this paper are collected in the following database record: biostudies:S-SCDT-10_1038-S44321-025-00293-5.

## Peer review information

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

## Acknowledgements

This work was supported by grants from the National Natural Science Foundation of China (No. 82173208, No.82473380), the Key Project of Science & Technology Development Fund of Tianjin Education Commission for Higher Education (No. 2022ZD064), the Scientific and Technological Projects of Tianjin (24ZXZSSS00050) and Tianjin Key Medical Discipline (Specialty) Constrction Project (TJYXZDXK-010A). We gratefully acknowledge the High-Throughput Screening Platform of Tianjin Medical University for their support. We thank Mingjie Chen (Shanghai NewCore Biotechnology Co., Ltd.) and Figdraw for providing data analysis and visualization support.

## Author contributions

**Yuanying Feng**: Data curation; Formal analysis; Investigation; Writing—original draft. **Yuchao He**: Data curation; Investigation. **Ran Zuo**: Data curation; Formal analysis. **Wenchen Gong**: Resources; Data curation. **Yuan Gao**: Data curation; Methodology. **Yun Wang**: Data curation. **Yu Wang**: Investigation. **Wenshuai Chen**: Methodology. **Liwei Chen**: Investigation. **Yi Luo**: Methodology. **Dongqi Yuan**: Data curation. **Peng Chen**: Conceptualization; Resources. **Hua Guo**: Conceptualization; Supervision; Funding acquisition; Investigation; Writing—review and editing.

Source data underlying figure panels in this paper may have individual authorship assigned. Where available, figure panel/source data authorship is listed in the following database record: biostudies:S-SCDT-10_1038-S44321-025-00293-5.

## Disclosure and competing interests statement

The authors declare no competing interests.

# Expanded View Figures

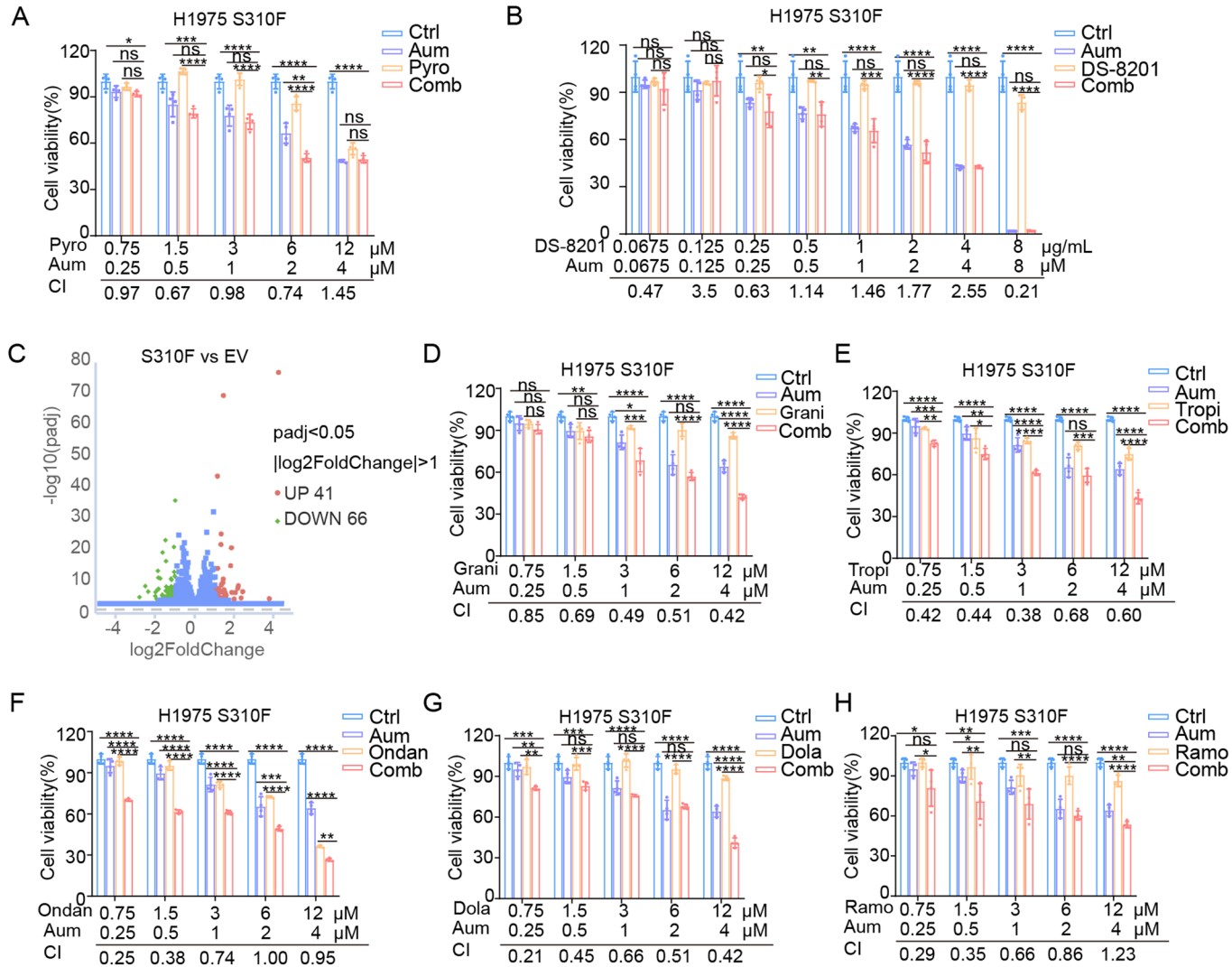

◀   **Figure EV1.   Therapeutic efficacy of aumolertinib combined with HER2-targeted agents and HTR3 antagonists.**

(A) Cell viability and CI analysis in H1975 S310F cells treated with indicated concentrations of aumolertinib and pyrotinib alone or in combination for 72 h ($n = 4$). $P$ value (Aum 0.25 μM, Pyro 0.75 μM): Ctrl vs Comb, $P = 0.0204$; $P$ value (Aum 0.5 μM, Pyro 1.5 μM): Ctrl vs Comb, $P = 0.0004$, Pyro vs Comb, $P < 0.0001$; $P$ value (Aum 1 μM, Pyro 3 μM): Ctrl vs Comb, $P < 0.0001$, Pyro vs Comb, $P < 0.0001$; $P$ value (Aum 2 μM, Pyro 6 μM): Ctrl vs Comb, $P < 0.0001$, Aum vs Comb, $P = 0.0037$, Pyro vs Comb, $P = 0.0001$; $P$ value (Aum 4 μM, Pyro 12 μM): Ctrl vs Comb, $P < 0.0001$. (B) Cell viability and CI analysis in H1975 S310F cells treated with indicated concentrations of aumolertinib and DS-8201 alone or in combination for 72 h ($n = 4$). $P$ value (Aum 0.25 μM, DS-8201 0.25 μg/ml): Ctrl vs Comb, $P = 0.007$, DS-8201 vs Comb, $P = 0.0239$; $P$ value (Aum 0.5 μM, DS-8201 0.5 μg/ml): Ctrl vs Comb, $P = 0.0011$, DS-8201 vs Comb, $P = 0.003$; $P$ value (Aum 1 μM, DS-8201 1 μg/ml): Ctrl vs Comb, $P < 0.0001$, DS-8201 vs Comb, $P = 0.0002$; $P$ value (Aum 2 μM, DS-8201 2 μg/ml): Ctrl vs Comb, $P < 0.0001$, DS-8201 vs Comb, $P < 0.0001$; $P$ value (Aum 4 μM, DS-8201 4 μg/ml): Ctrl vs Comb, $P < 0.0001$, DS-8201 vs Comb, $P < 0.0001$; $P$ value (Aum 8 μM, DS-8201 8 μg/ml): Ctrl vs Comb, $P < 0.0001$, DS-8201 vs Comb, $P < 0.0001$. (C) Differential gene expression volcano plot of H1975 EV/S310F transcriptome sequencing. (D) Cell viability and CI analysis in H1975 S310F cells treated with indicated concentrations of aumolertinib and granisetron alone or in combination for 72 h ($n = 4$). $P$ value (Aum 0.5 μM, Grani 1.5 μM): Ctrl vs Comb, $P = 0.0050$; $P$ value (Aum 1 μM, Grani 3 μM): Ctrl vs Comb, $P < 0.0001$, Aum vs Comb, $P = 0.0208$, Grani vs Comb, $P = 0.0002$; $P$ value (Aum 2 μM, Grani 6 μM): Ctrl vs Comb, $P < 0.0001$, Grani vs Comb, $P < 0.0001$; $P$ value (Aum 4 μM, Grani 12 μM): Ctrl vs Comb, $P < 0.0001$, Aum vs Comb, $P < 0.0001$, Grani vs Comb, $P < 0.0001$. (E) Cell viability and CI analysis in H1975 S310F cells treated with indicated concentrations of aumolertinib and tropisetron alone or in combination for 72 h ($n = 4$). $P$ value (Aum 0.25 μM, Tropi 0.75 μM): Ctrl vs Comb, $P < 0.0001$, Aum vs Comb, $P = 0.0003$, Tropi vs Comb, $P = 0.0012$; $P$ value (Aum 0.5 μM, Tropi 1.5 μM): Ctrl vs Comb, $P < 0.0001$, Aum vs Comb, $P = 0.0050$, Tropi vs Comb, $P = 0.0270$; $P$ value (Aum 1 μM, Tropi 3 μM): Ctrl vs Comb, $P < 0.0001$, Aum vs Comb, $P < 0.0001$, Tropi vs Comb, $P < 0.0001$; $P$ value (Aum 2 μM, Tropi 6 μM): Ctrl vs Comb, $P < 0.0001$, Tropi vs Comb, $P = 0.0001$; $P$ value (Aum 4 μM, Tropi 12 μM): Ctrl vs Comb, $P < 0.0001$, Aum vs Comb, $P < 0.0001$, Tropi vs Comb, $P < 0.0001$. (F) Cell viability and CI analysis in H1975 S310F cells treated with indicated concentrations of aumolertinib and ondansetron alone or in combination for 72 h ($n = 4$). $P$ value (Aum 0.25 μM, Ondan 0.75 μM): Ctrl vs Comb, $P < 0.0001$, Aum vs Comb, $P < 0.0001$, Ondan vs Comb, $P < 0.0001$; $P$ value (Aum 0.5 μM, Ondan 1.5 μM): Ctrl vs Comb, $P < 0.0001$, Aum vs Comb, $P < 0.0001$, Ondan vs Comb, $P < 0.0001$; $P$ value (Aum 1 μM, Ondan 3 μM): Ctrl vs Comb, $P < 0.0001$, Aum vs Comb, $P < 0.0001$, Ondan vs Comb, $P < 0.0001$; $P$ value (Aum 2 μM, Ondan 6 μM): Ctrl vs Comb, $P < 0.0001$, Aum vs Comb, $P = 0.0006$, Ondan vs Comb, $P < 0.0001$; $P$ value (Aum 4 μM, Ondan 12 μM): Ctrl vs Comb, $P < 0.0001$, Aum vs Comb, $P < 0.0001$, Ondan vs Comb, $P = 0.0020$. (G) Cell viability and CI analysis in H1975 S310F cells treated with indicated concentrations of aumolertinib and dolasetron alone or in combination for 72 h ($n = 4$). $P$ value (Aum 0.25 μM, Dola 0.75 μM): Ctrl vs Comb, $P = 0.0003$, Aum vs Comb, $P = 0.0039$, Dola vs Comb, $P = 0.0015$; $P$ value (Aum 0.5 μM, Dola 1.5 μM): Ctrl vs Comb, $P = 0.0007$, Dola vs Comb, $P = 0.0009$; $P$ value (Aum 1 μM, Dola 3 μM): Ctrl vs Comb, $P < 0.0001$, Dola vs Comb, $P < 0.0001$; $P$ value (Aum 2 μM, Dola 6 μM): Ctrl vs Comb, $P < 0.0001$, Dola vs Comb, $P < 0.0001$; $P$ value (Aum 4 μM, Dola 12 μM): Ctrl vs Comb, $P < 0.0001$, Aum vs Comb, $P < 0.0001$, Dola vs Comb, $P < 0.0001$. (H) Cell viability and CI analysis in H1975 S310F cells treated with indicated concentrations of aumolertinib and ramosetron alone or in combination for 72 h ($n = 4$). $P$ value (Aum 0.25 μM, Ramo 0.75 μM): Ctrl vs Comb, $P = 0.0179$, Ramo vs Comb, $P = 0.0174$; $P$ value (Aum 0.5 μM, Ramo 1.5 μM): Ctrl vs Comb, $P = 0.0023$, Aum vs Comb, $P = 0.0440$, Ramo vs Comb, $P = 0.0059$; $P$ value (Aum 1 μM, Ramo 3 μM): Ctrl vs Comb, $P = 0.0004$, Ramo vs Comb, $P = 0.0082$; $P$ value (Aum 2 μM, Ramo 6 μM): Ctrl vs Comb, $P < 0.0001$, Ramo vs Comb, $P < 0.0001$; $P$ value (Aum 4 μM, Ramo 12 μM): Ctrl vs Comb, $P < 0.0001$, Aum vs Comb, $P = 0.0074$, Ramo vs Comb, $P < 0.0001$. Data are presented as mean ± SD. Statistical test: one-way ANOVA, ns: not significant, *$P < 0.05$, **$P < 0.01$, ***$P < 0.001$, and ****$P < 0.0001$.

   

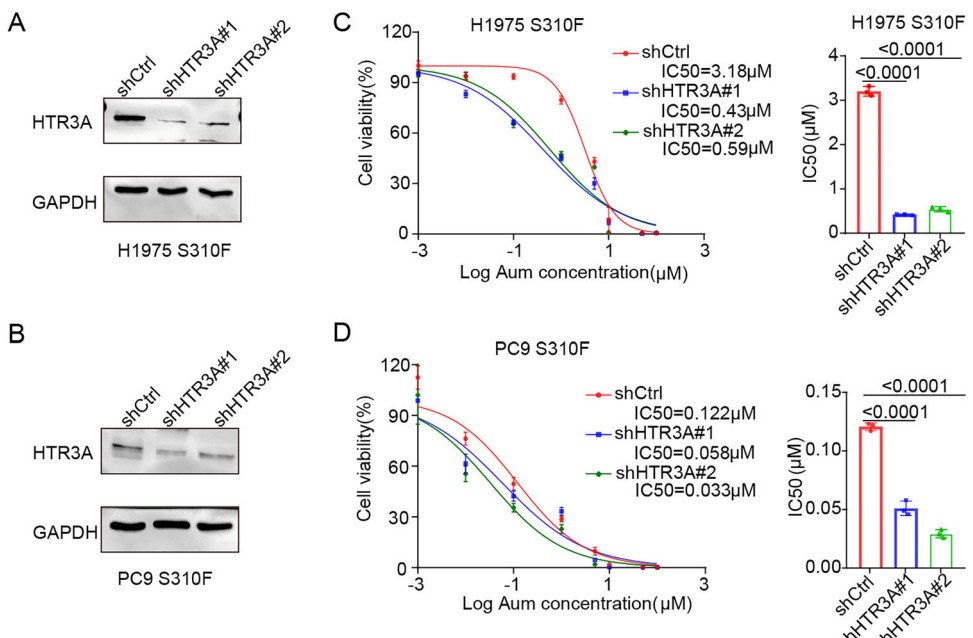

**Figure EV2.   HTR3A knockdown potentiates the sensitivity to aumolertinib in HER2 S310F-mutant cells.**

(A, B) Western blot validation of stable HTR3A knockdown cell lines. (C, D) The aumolertinib $IC_{50}$ value of H1975 S310F and PC9 S310F cells were calculated between shCtrl and shHTR3A groups after treating with aumolertinib at the indicated concentrations for 5 days ($n = 3$). Data are presented as mean ± SD. Statistical test: two-tailed unpaired Student's $t$ test.

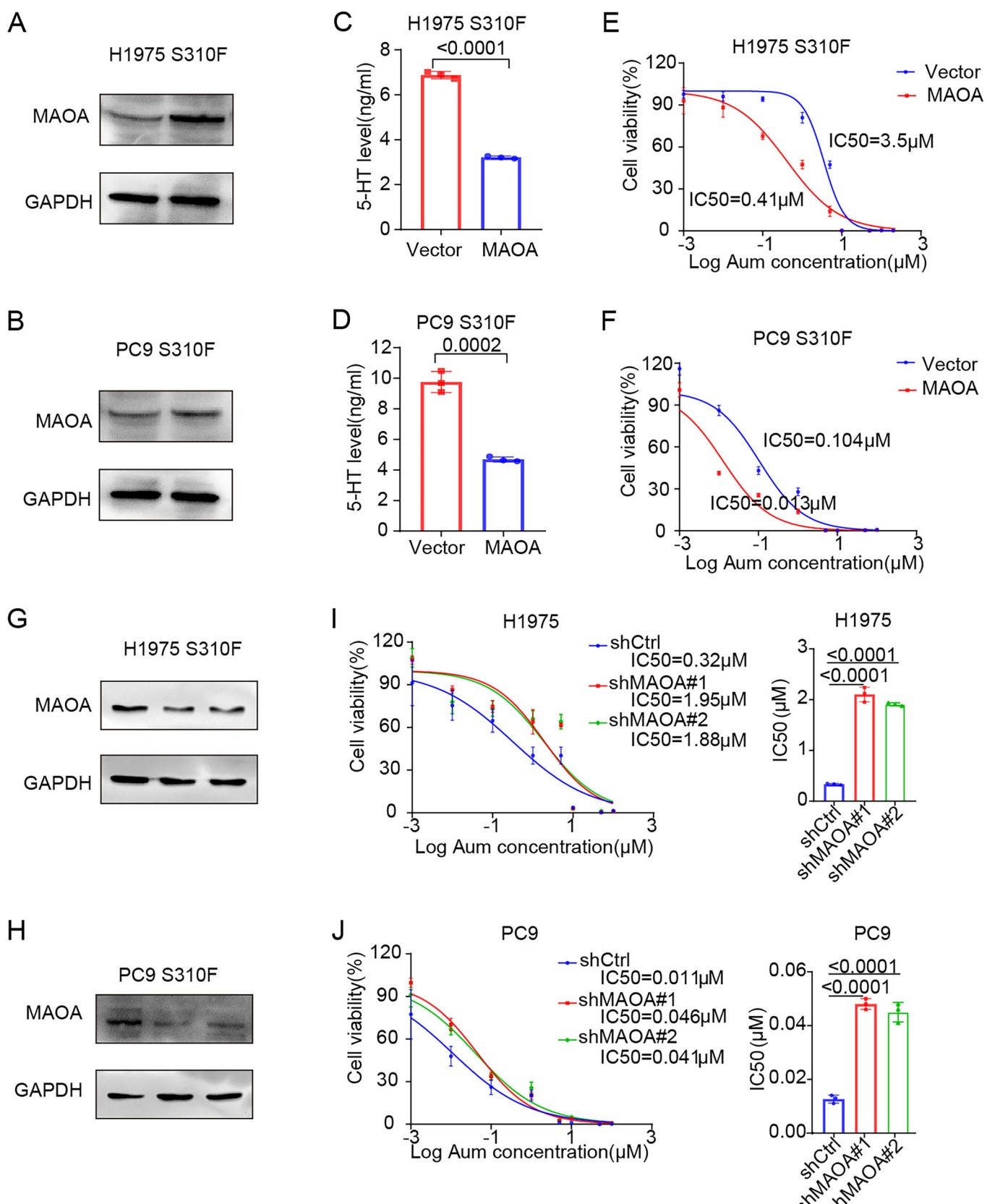

**Figure EV3. MAOA expression regulates aumolertinib resistance.**

(**A, B**) H1975 S310F and PC9 S310F cells stably overexpressing MAOA were validated by western blot. (**C, D**) 5-HT levels in the supernatant ($n = 3$). (**E, F**) The aumolertinib $IC_{50}$ value of H1975 S310F and PC9 S310F cells were calculated between vector and MAOA overexpression groups after treating with aumolertinib at the indicated concentrations for 5 days ($n = 4$). (**G, H**) Western blot validation of stable MAOA knockdown cell lines. (**I, J**) The aumolertinib $IC_{50}$ value of H1975 and PC9 cells was calculated between shCtrl and shMAOA groups after treating with aumolertinib at the indicated concentrations for 5 days ($n = 3$). Data are presented as mean ± SD. Statistical test: Two-tailed unpaired Student's $t$ test.

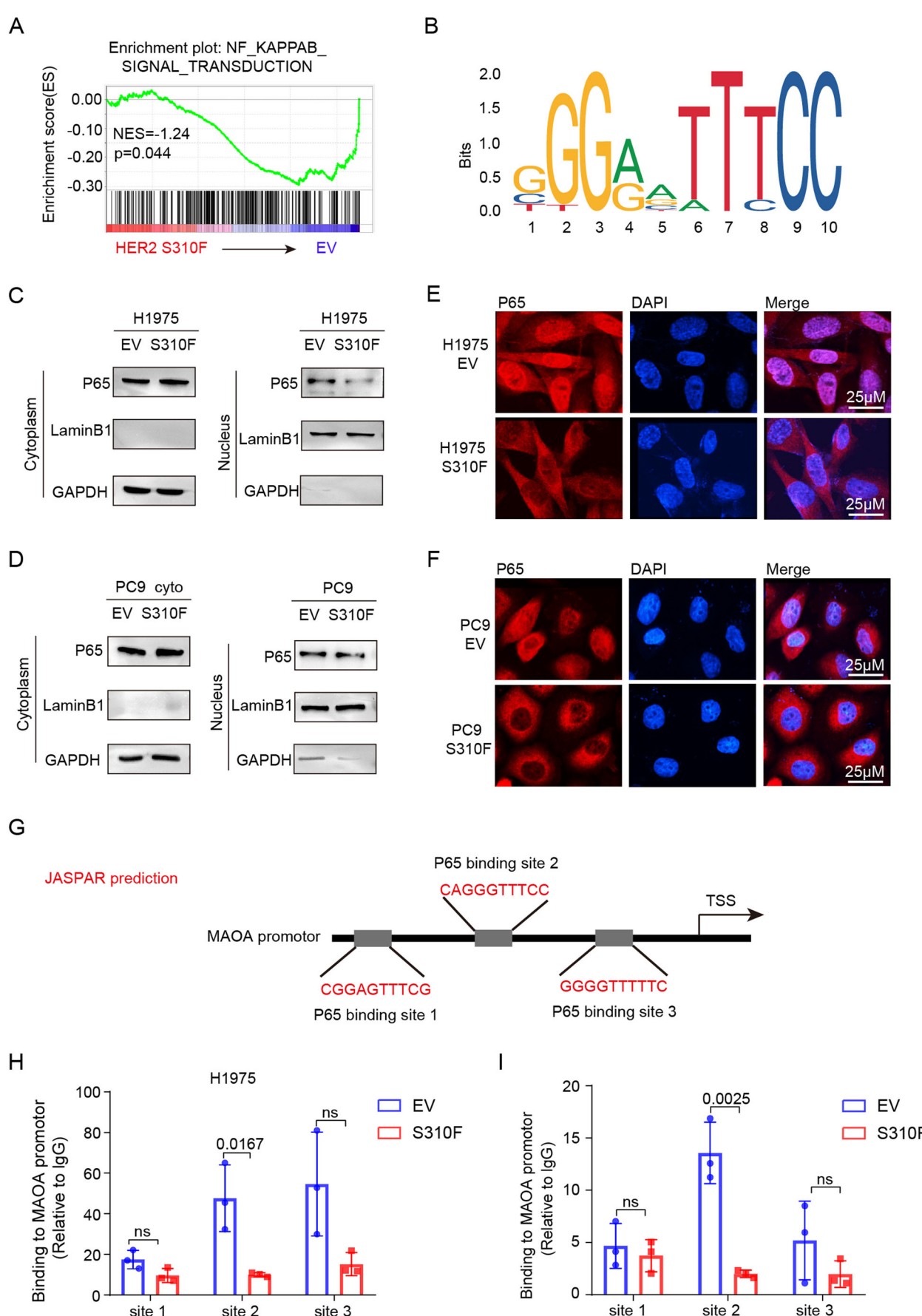

**Figure EV4.   The mechanism of downregulation of MAOA expression.**

(A) Perform GSEA analysis on H1975 EV/S310F RNA seq data. (B) The JASPAR database forecasts the sequence of the P65 binding site within the promoter region of the MAOA gene. (C, D) The nuclear and cytoplasmic expression levels of P65 were assessed in H1975 EV/S310F and PC9 EV/S310F cells through nuclear–cytoplasmic separation experiments. (E, F) Immunofluorescence assays were conducted to determine the localization of P65 in H1975 EV/S310F and PC9 EV/S310F cells, with P65 appearing in red and the nuclei in blue. (G) The P65-binding motif was predicted by JASPAR, and schematic images of the potential P65 binding sites in the MAOA promoter region are shown. (H, I) ChIP-qPCR analysis of P65 binding on MAOA promoter in H1975 EV/S310F and PC9 EV/S310F cells ($n = 3$). Data are presented as mean ± SD. Statistical test: Two-tailed unpaired Student's $t$ test.

