## [Peer Review File · EMBO Molecular Medicine]

5-HT regulates resistance to aumolertinib by attenuating ferroptosis in lung adenocarcinoma

Hua Guo, Yuanying Feng, Yuchao He, Ran Zuo, Wenchen Gong, Yuan Gao, Yun Wang, Yu Wang, Wenshuai Chen, Liwei Chen, Yi Luo, Dongqi Yuan, and Peng Chen

Corresponding authors: Hua Guo (guohua@tmu.edu.cn) , Peng Chen (chenpeng@tjmuch.com)

Review Timeline:

Submission Date:	5th Mar 25
Editorial Decision:	4th Apr 25
Revision Received:	19th Jun 25
Editorial Decision:	2nd Jul 25
Revision Received:	12th Jul 25
Accepted:	25th Jul 25

Editor: Lise Roth

Transaction Report:

4th Apr 2025

Dear Prof. Guo,

Thank you for the submission of your manuscript to EMBO Molecular Medicine. We have now received feedback from the three reviewers who agreed to evaluate your manuscript. As you will see from the reports below, the referees acknowledge the interest of the study and are overall supporting publication of your work pending appropriate revisions.

Addressing the reviewers' concerns in full will be necessary for further considering the manuscript in our journal, and acceptance of the manuscript will entail a second round of review. However, please note that we do NOT ask for validation in patient-derived models or transgenic mice (referee #1 point #6).

EMBO Molecular Medicine encourages a single round of revision only and therefore, acceptance or rejection of the manuscript will depend on the completeness of your responses included in the next, final version of the manuscript. For this reason, and to save you frustration at the end, I would strongly discourage you from returning an incomplete revision.

We are expecting your revised manuscript within three to four months, if you anticipate any delay, please contact us.

We require:

4) A .docx formatted letter INCLUDING the reviewers' reports and your detailed point-by-point responses to their comments. As part of the EMBO Press transparent editorial process, the point-by-point response is part of the Review Process File (RPF), which will be published alongside your paper.

5) A complete author checklist, which you can download from our author guidelines (<https://www.embopress.org/page/journal/17574684/authorguide#submissionofrevisions>). Please insert information in the checklist that is also reflected in the manuscript. The completed author checklist will also be part of the RPF.

6) All Materials and Methods need to be described in the main text using our 'Structured Methods' format. According to this format, the Methods section includes a Reagents and Tools Table (listing key reagents, experimental models, software and relevant equipment and including their sources and relevant identifiers) followed by a Methods and Protocols section describing the methods, ideally using a step-by-step protocol format. The aim is to facilitate adoption of the methodologies across labs. Please download and fill our Reagents and Tools Table template (.docx), which you can find in our author guidelines: <https://www.embopress.org/page/journal/14693178/authorguide#structuredmethods>.

<https://www.embopress.org/doi/10.15252/msb.20178071>

7) Please note that all corresponding authors are required to supply an ORCID ID for their name upon submission of a revised manuscript.

8) It is mandatory to include a 'Data Availability' section after the Materials and Methods. Before submitting your revision, primary datasets produced in this study need to be deposited in an appropriate public database, and the accession numbers and database listed under 'Data Availability'. Please remember to provide a reviewer password if the datasets are not yet public (see

<https://www.embopress.org/page/journal/17574684/authorguide#dataavailability>).

9) For data quantification: please specify the name of the statistical test used to generate error bars and P values, the number (n) of independent experiments (specify technical or biological replicates) underlying each data point and the test used to calculate p-values in each figure legend. The figure legends should contain a basic description of n, P and the test applied. Graphs must include a description of the bars and the error bars (s.d., s.e.m.). Please provide exact p values.

10) Our journal encourages inclusion of *data citations in the reference list* to directly cite datasets that were re-used and obtained from public databases. Data citations in the article text are distinct from normal bibliographical citations and should directly link to the database records from which the data can be accessed. In the main text, data citations are formatted as follows: "Data ref: Smith et al, 2001" or "Data ref: NCBI Sequence Read Archive PRJNA342805, 2017". In the Reference list, data citations must be labeled with "[DATASET]". A data reference must provide the database name, accession number/identifiers and a resolvable link to the landing page from which the data can be accessed at the end of the reference. Further instructions are available at .

11) We replaced Supplementary Information with Expanded View (EV) Figures and Tables that are collapsible/expandable online. EV Figures should be cited as 'Figure EV1, Figure EV2' etc... in the text and their respective legends should be included in the main text after the legends of regular figures.

12) The paper explained: EMBO Molecular Medicine articles are accompanied by a summary of the articles to emphasize the major findings in the paper and their medical implications for the non-specialist reader. Please provide a draft summary of your article highlighting

13) Author contributions: CRedit has replaced the traditional author contributions section because it offers a systematic machine readable author contributions format that allows for more effective research assessment. Please remove the Authors Contributions from the manuscript and use the free text boxes beneath each contributing author's name in our system to add specific details on the author's contribution. More information is available in our guide to authors.

Please also suggest a visual abstract to illustrate your article as a PNG file 550 px wide x 300-600 px high. A cropped portion of this image will serve as thumbnail for the table of content on our webpage.

16) As part of the EMBO Publications transparent editorial process initiative (see our Editorial at <http://embomolmed.embopress.org/content/2/9/329>), EMBO Molecular Medicine will publish online a Review Process File (RPF) to accompany accepted manuscripts.

In the event of acceptance, this file will be published in conjunction with your paper and will include the anonymous referee reports, your point-by-point response and all pertinent correspondence relating to the manuscript. Let us know whether you agree with the publication of the RPF and as here, if you want to remove or not any figures from it prior to publication.

I look forward to receiving your revised manuscript.

Yours sincerely,

Lise Roth

***** Reviewer's comments *****

Referee #1 (Comments on Novelty/Model System for Author):

The study effectively establishes a link between the HER2 S310F mutation and resistance to third-generation EGFR-TKIs, which is highly relevant for precision oncology. The study utilizes a combination of in vitro (cell lines) and in vivo (xenograft) models, enhancing the conclusions' robustness and use of high-throughput screening (HTS) to identify sensitizers is a strong methodological approach. This strategy not only enhances our understanding but also provides important mechanistic insights. Identifying 5-HT accumulation as a driver of aumolertinib resistance is novel and offers a unique angle. The identification of HTR3 antagonists as potential sensitizers for aumolertinib is promising and the in vivo data demonstrating the efficacy and safety of the aumolertinib/palonosetron combination provides strong preclinical support for further investigation.

Referee #1 (Remarks for Author):

Remarks for Author:

The study effectively establishes a link between the HER2 S310F mutation and resistance to third-generation EGFR-TKIs, which is highly relevant for precision oncology. The study utilizes a combination of in vitro (cell lines) and in vivo (xenograft) models, enhancing the conclusions' robustness and use of high-throughput screening (HTS) to identify sensitizers is a strong methodological approach. The inclusion of transcriptomic analysis methods such as RNA-seq, cMAP, KEGG, and GSEA to investigate resistance mechanisms is a valuable approach. This strategy not only enhances our understanding but also provides important mechanistic insights, and the identification of 5-HT accumulation as a driver of aumolertinib resistance is novel and provides a unique angle. The identification of HTR3 antagonists as potential sensitizers for aumolertinib is promising and the in vivo data demonstrating the efficacy and safety of the aumolertinib/palonosetron combination provides strong preclinical support for further investigation. But some issues need to be addressed.

Major comments:

- 1) The connection between HER2 S310F, MAOA downregulation, and NF- κ B signaling is intriguing but could benefit from additional validation (e.g., chromatin immunoprecipitation for P65 binding).
- 2) The role of Ca²⁺/CAMKK2/AMPK in ferroptosis regulation is well-supported, but alternative pathways (e.g., autophagy) could be explored for further validation.
- 3) Rescue experiments with MAOA reconstitution or CRISPR knockout of HTR3 could further validate key mechanisms.
- 4) Detailed toxicity assessment: While body weight and organ function markers were monitored, histopathological analysis of major organs could provide a more comprehensive safety profile.
- 5) Please describe how the H1975 aumolertinib secondary resistance cells (H1975 AR) were established, and do these cells

have HER2 S310F mutant?

6) Overall, the study presents a compelling case for HER2 S310F-driven EGFR-TKI resistance and highlights a potential therapeutic strategy with HTR3 antagonists. Further validation, particularly in patient-derived models or transgenic mice could pave the way for clinical translation.

7) Are other third-generation TKI (e.g., Osimertinib) also through 5-HT regulated resistance to HER2 S310F-driven EGFR-TKI resistance?

Minor comments:

1) The X axis of Fig. 2G "Alm" should be "Aum".

2) The two marked lines in Fig. 7C were converse.

Referee #2 (Comments on Novelty/Model System for Author):

5-HT regulates resistance to aumolertinib by attenuating ferroptosis in lung adenocarcinoma

Feng et al. applied an integrative FDA-approved library screening and transcriptomic analysis to evaluate novel therapeutic approaches for EGFR-TKI-resistant lung adenocarcinoma (LUAD) patients. They identified HTR3 antagonists as a promising drug, which mechanism involves resistance to ferroptosis by accumulation of 5-HT, therefore triggering calcium influx and activating Ca²⁺/CAMKK2/AMPK signaling, ultimately inducing aumolertinib resistance. This study opens a broad and promising perspective for overcoming the clinically challenging 3rd generation EGFR-TKI resistance, as well as a potential early LUAD diagnosis by direct measurement of 5-HT levels. Moreover, their contribution is relevant for the Pneumology and Cellular Biology fields, it is well structured, and their results clearly presented. However, I would like to suggest several points in order to improve the current version of the manuscript:

MINOR CONCERNS

1. Table EV1 should also include cancer stages (if possible at initial and final diagnosis), are all 6 patients with co-mutations in advanced stage? This information should also be added to Fig. 1A.
2. To complete the mutational landscape, what is the status of KRAS in selected patients, since mutations on both KRAS and EGFR driver oncogenes have been associated to chemotherapy resistance (for instance, include PMID: 38023198, 27055085)?
3. Importantly, therapy-induced EGFR mutations also induce phenotypic transformation from NCSLC to SCLC. HER2 mutations have been reported in SCLC patients. Therefore, what could be the implication of this mechanism in the suggested model, can authors discard histologic alterations during acquired resistance? (Revise PMID: 34729945, 37215577).
4. Include in Discussion additional evidence for HTR-targeted pharmacology in cancer (for instance, include PMID 38037454). Previous groups have highlighted ferroptosis resistance implications in other cancer subtypes.
5. Fig. EV6B shows a single results from the JASPAR prediction, without statistical values of % of targets including this motif. Other authors have shown a FOXH1 recruitment in a HTR3A-dependent manner (PMID: 39046651). Is this consistent with their own data? They could show a more detailed analysis.
6. Include p65 and its suggested genomic binding effects on MAOA in their Graphical Abstract in Fig. 8. On the contrary, NRF2 plays a crucial role in the suggested model, however this work lacks robust evidence showing a causal involvement of NRF2 by loss-of-function experiments.
7. "Ferroptosis-related indicators" is a better term instead of "biomarkers, -related genes, effectors"?
8. Describe in more detailed the procedure for the establishment of H1975 AR cells.
9. Prepare a Supplementary Material file with uncropped WB images.

Referee #3 (Remarks for Author):

Thank you for sharing your insightful work on the mechanisms of aumolertinib resistance. I have a few suggestions that may further strengthen the study and clarify the proposed mechanisms:

-In Figure 2D, you demonstrate that HTR3 antagonists in combination with aumolertinib synergistically inhibit cell viability. However, to rule out potential off-target effects of these small molecules, it would be valuable to perform HTR3-specific knockdown (e.g., via siRNA, shRNA, or CRISPR/Cas9). If silencing HTR3 recapitulates the combination effect seen with the pharmacological inhibitors, it will strongly support that the observed synergy truly stems from HTR3 blockade rather than other off-target activities.

-In Figure 2B, ALK inhibitors were also among the candidate agents identified. I wonder why ALK inhibitors were also effective in

overcoming the resistance.

-You demonstrate in Figure 3E that 5-HT levels are notably elevated in resistant cells. It would be interesting to examine whether reducing 5-HT itself (e.g., via inhibition of its synthesis or by enhancing its metabolism) can restore sensitivity to aumolertinib. Such an experiment would provide direct causal evidence that 5-HT accumulation drives resistance.

-Your data highlight MAOA's pivotal role in modulating 5-HT levels and resistance. You show that overexpression of MAOA can reverse resistance in resistant cells. As a complementary approach, it would be informative to knock down or knock out MAOA in parental cells (e.g., H1975, PC9, H3255) to see if such depletion confers or enhances resistance. This loss-of-function experiment would strongly reinforce the model that reduced MAOA is a central driver of the resistance phenotype.

-Since MAOA plays key roles in various biological processes, including mood regulation and tumor biology, there may be previous studies or case reports describing MAOA mutations or altered expression in the context of cancer drug resistance. Including a brief literature overview or discussion of MAOA's known roles in drug resistance could help frame your findings in a broader context.

Overall, your study provides a compelling rationale for targeting the 5-HT/HTR3 axis to combat aumolertinib resistance, particularly in the setting of HER2 S310F-mutant lung adenocarcinoma. Addressing the above points may further strengthen your conclusions and expand the potential impact of your work.

Response to reviewers' comments

Dear Editors and Reviewers,

Thanks very much for taking your time to review this manuscript. We write to submit the revised manuscript titled "5-HT regulates resistance to aumolertinib by attenuating ferroptosis in lung adenocarcinoma" (ID: EMM-2025-21570-V2) to *EMBO Molecular Medicine*. The comments from the editor and reviewers are very helpful for us to improve the quality and significance of the manuscript. Here, we performed considerable changes and presented new experimental data and progress. We did additional experiments (Fig. EV2, Fig EV3G-J, Fig EV4G-I, Fig S2, Fig S4, Fig S5, Fig S6, Fig S7) and revised the text accordingly. We highlighted the changes in yellow in the revised manuscript. As a result, we believe the revision manuscript is much stronger. Point by point responses to the reviewers' comments are listed below. Thank you again for your positive comments and valuable suggestions to improve the quality of our manuscript.

Comments from Reviewer #1

Q1: The connection between HER2 S310F, MAOA downregulation, and NF- κ B signaling is intriguing but could benefit from additional validation (e.g., chromatin immunoprecipitation for P65 binding).

Answer: We appreciate you taking the time to read our text and your insightful feedback. We have confirmed that HER2 S310F mutation induces the reduced expression of MAOA, and further investigation revealed diminished P65 nuclear translocation, suggesting impaired P65 binding to the MAOA promoter as a potential mechanism contributing to reduced MAOA expression in *HER2* S310F-mutant cells (Fig 4C-F, Fig EV4A-E). Nevertheless, we previously lack direct experimental evidence to substantiate this hypothesis. As suggested, we designed primers targeting three predicted P65 binding sites within the MAOA promoter region (identified via JASPAR), which were validated by ChIP-qPCR (Fig. EV4G). This analysis confirmed specific binding of P65 to Site 2 of the MAOA promoter (Fig. EV4H-4I). These data suggest that diminished P65 binding at MAOA promoter Site 2 in *HER2* S310F mutant cells suppresses MAOA transcription. These new data have been added to the revised version of the manuscript in yellow highlight.

Fig EV4. G) The P65-binding motif was predicted by JASPAR, and schematic images of the potential P65 binding sites in the MAOA promoter region are shown. H, I) ChIP-qPCR analysis of P65 binding on MAOA promoter in H1975 EV/S310F and PC9 EV/S310F cells (n = 3). Data are presented as mean \pm SD. Statistical test: Two-tailed unpaired Student's t test.

Q2: *The role of Ca²⁺/CAMKK2/AMPK in ferroptosis regulation is well-supported, but alternative pathways (e.g., autophagy) could be explored for further validation.*

Answer: Thank you for your suggestion. Our data demonstrated that 5-HT/palonosetron modulates ferroptosis through the Ca²⁺/CAMKK/AMPK axis (Fig. 5). While prior studies implicate autophagy and other pathways in ferroptosis regulation, consistent with your suggestion (Zhou et al, 2024). We followed your suggestion and validated autophagic markers after 5-HT/palonosetron treatment, but the result revealed no detectable alterations (Appendix Figure S4.). Nevertheless, potential involvement of alternative ferroptosis-related pathways, such as TP53 signaling or mTOR, in 5-HT-mediated ferroptosis regulation warrants future investigation. These new data have been added to the revised version of the manuscript in yellow highlight.

Appendix Figure S4 Western blotting analysis was performed to assess the modulatory effects of 5-HT and palonosetron on core autophagy markers.

Q3: *Rescue experiments with MAOA reconstitution or CRISPR knockout of HTR3 could further validate key mechanisms.*

Answer: Thank you for your valuable suggestion to make our article more complete. We found that among genes associated with 5-HT synthesis, transport, and metabolism, only the 5-HT metabolizing enzyme MAOA was significantly downregulation in HER2 S310F cells (Fig. 4). As suggested, to further investigate the impact of MAOA expression on 5-HT levels and Aumolertinib sensitivity, we performed rescue experiments by overexpressing MAOA in H1975 S310F and PC9 S310F cells. MAOA overexpression significantly reduced extracellular 5-HT levels and decreased the IC50 of Aumolertinib (Fig EV3A-F). Conversely, knockdown of MAOA in wild-type H1975 and PC9 cells significantly increased the IC50 of Aumolertinib (Fig EV3G-J). Collectively, these results establish a critical role for MAOA in regulating 5-HT metabolism and modulating cellular sensitivity to Aumolertinib.

Our previous results demonstrated that HTR3 antagonists significantly reduced the IC50 of Aumolertinib (Fig 2G, 2I). To further define the role of HTR3 in this effect and rule out potential off-target effects of the pharmacological inhibitors, we generated stable HTR3-knockdown cell lines. Subsequent CCK8 experiments confirmed that HTR3 knockdown similarly reduced the IC50 of Aumolertinib (Fig EV2A-D). This genetic evidence confirms that the reduction in Aumolertinib IC50 by HTR3 antagonists is attributable to on-target HTR3 inhibition, rather than off-target effects. These new data have been added to the revised version of the manuscript in yellow highlight.

Fig EV3 MAOA expression regulate aumolertinib resistance. A, D) H1975 S310F and PC9 S310F cells stably overexpressing MAOA were validated by Western blot. B, E) 5-HT levels in the supernatant (n = 3). C, F) The aumolertinib IC₅₀ value of H1975 S310F and PC9 S310F cells were calculated between vector and MAOA overexpression groups after treating with almonertinib at the indicated concentrations for 5 days. G, I) Western blot validation of stable MAOA knockdown cell lines. H, J) The aumolertinib IC₅₀ value of H1975 and PC9 cells were calculated between shCtrl and shMAOA groups after treating with almonertinib at the indicated concentrations for 5 days. Data are presented as mean ± SD. Statistical test: Two-tailed unpaired Student's t test.

Fig EV2 HTR3A knockdown potentiates the sensitivity to aumolertinib in HER2 S310F-mutant cells. A, B) Western

blot validation of stable HTR3A knockdown cell lines. C, D) The aumolertinib IC50 value of H1975 S310F and PC9 S310F cells were calculated between shCtrl and shHTR3A groups after treating with almonertinib at the indicated concentrations for 5 days.

Q4: *Detailed toxicity assessment: While body weight and organ function markers were monitored, histopathological analysis of major organs could provide a more comprehensive safety profile.*

Answer: Thank you for your valuable suggestion. As recommended by the reviewer, hematoxylin and eosin (H&E) staining was performed on cardiac, hepatic, and renal tissues from control mice and those administered aumolertinib monotherapy, palonosetron monotherapy, or combination therapy. Histopathological analysis revealed no evident morphological alterations in any treatment cohort compared to controls (Appendix Figure S6). These findings corroborate the absence of overt toxicological effects associated with the combination of aumolertinib and palonosetron. These new data have been added to the revised version of the manuscript in yellow highlight.

Appendix Figure S6 Histopathological analysis of murine cardiac, hepatic, and renal tissues following aumolertinib, palonosetron monotherapy or combination therapy.

Q5: *Please describe how the H1975 aumolertinib secondary resistance cells (H1975 AR) were established, and do these cells have HER2 S310F mutant?*

Answer: Thank you for your valuable suggestion. H1975 aumolertinib secondary resistance cells were generated from parental H1975 cells via stepwise dose escalation over 24 weeks: beginning at 10 nM in week 2 and progressively increasing to a final concentration of 10 μ M aumolertinib, as previously reported(Yue et al, 2025). We have incorporated this content with yellow highlight in the Methods.

Q6: Overall, the study presents a compelling case for HER2 S310F-driven EGFR-TKI resistance and highlights a potential therapeutic strategy with HTR3 antagonists. Further validation, particularly in patient-derived models or transgenic mice could pave the way for clinical translation.

Answer: Thank you for your valuable suggestion. We concur with the reviewer that incorporating patient-derived models or transgenic mice would enhance clinical relevance. Patient-derived models (PDMs) comprised of patient-derived xenografts (PDXs), *in vitro* patient-derived tumor cell cultures (PDCs) as well as patient-derived organoids (PDO). NSCLC patient-derived organoids retain both histopathological features of parental tumors and their sensitivity to targeted therapies, thereby enabling validation and discovery of biomarker-therapeutic associations(Shi et al, 2020). We collected postoperative tissues from 3 patients with lung adenocarcinoma and performed organoid culture. Following 10 days of culture, organoids were divided into four groups: vehicle control, aumolertinib monotherapy, aumolertinib + 5-HT, and aumolertinib + 5-HT + palonosetron. After 72 hours of drug treatment, the CCK8 test results showed that 5-HT significantly attenuated the cytotoxic efficacy of aumolertinib in patient-derived lung cancer organoids, while the addition of palonosetron treatment counteracted the effect of 5-HT (Appendix Figure S7). This finding is consistent with results from both *in vitro* cellular experiments and *in vivo* nude mouse models, corroborating that elevated 5-HT levels drive Aumolertinib resistance. Critically, HTR3 antagonists reversed 5-HT-mediated resistance to Aumolertinib, thereby enhancing the translational relevance of these results. These new data have been added to the revised version of the manuscript in yellow highlight.

Appendix Figure S7 Patient derived organoid validation of the effect of 5-HT/palonosetron on aumolertinib sensitivity. A-C) Viability of lung adenocarcinoma organoids after treatment with aumolertinib, 5-HT + aumolertinib or a combination of 5-HT, aumolertinib and palonosetron. Data are presented as mean \pm SD. Statistical test: one way ANOVA.

Q7: Are other third-generation TKI (e.g., Osimertinib) also through 5-HT regulated resistance to HER2 S310F-driven EGFR-TKI resistance?

Answer: Thank you for your suggestion to investigate whether 5-HT mediates osimertinib resistance. Initially, we established that the HER2 S310F mutation confers resistance to third-generation EGFR-TKI, Aumolertinib and Osimertinib (Fig 1C-E, Appendix Figure S1C-E). Subsequent mechanistic investigation focused on Aumolertinib as a representative agent. As suggested, we further investigated whether 5-HT similarly mediates resistance to Osimertinib. Exogenous 5-HT supplementation significantly increased the IC₅₀ of Osimertinib, phenocopying the resistance observed with Aumolertinib (Appendix Figure S2 A-C). Collectively, these findings demonstrate that elevated 5-HT levels mediate resistance to both Aumolertinib and Osimertinib. However, whether 5-HT represents a common resistance mechanism across all third-generation EGFR TKIs requires further validation. These new data have been added to the revised version of the manuscript in yellow highlight.

Appendix Figure S2 A-C) Dose-response curves determined by the CCK-8 assay were used to calculate the IC₅₀ values of osimertinib for 5 days in the presence or absence of 2.5 μ M 5-HT.

Minor comments:

1) The X axis of Fig. 2G "Alm" should be "Aum".

Answer: We sincerely thank you for pointing out the writing errors in our figures. We have corrected the identified writing errors and conducted a thorough review of the manuscript to ensure accuracy throughout.

2) The two marked lines in Fig. 7C were converse.

Answer: We sincerely thank you for pointing out the labeling errors in our figures. We have corrected the two marked lines in Fig. 7C and conducted a thorough review of the manuscript to ensure accuracy throughout.

Comments from Reviewer #2

***Q1:** Table EV1 should also include cancer stages (if possible at initial and final diagnosis), are all 6 patients with co-mutations in advanced stage? This information should also be added to Fig. 1A.*

Answer: We appreciate you taking the time to read our text and your insightful feedback. In response to this comment, we have revised Table EV1 to include clinical stages of initial diagnosis. However, the absence of post-treatment follow-up data precluded documentation of tumor staging at final diagnosis. Future studies will analyze treatment and prognosis in this HER2-altered cohort, with comprehensive TNM staging data acquisition. We have incorporated clinical stage in Fig 1A, as suggested.

Table EV1 Statistics on patients with HER2 alterations

HER2 Alteration	Subtype	N	EGFR Mutation	Cancer Stages				
				Stage	Stage	Stage	Stage	NA
				1	2	3	4	
Mutation	p. Y772-A775dup	30	0	13			16	1
	p. S310F/Y	7	6				7	
	p. G776delinsVC /LC	6	0	1		1	4	
	p. L755S/P	4	1	1	1		2	
	p. G778-P780dup	2	0				2	
	p. V842I	2	1	1			1	
	p. V659E	2	0		1		1	
	p. A446V	2	0		1			1
	p. V777L	1	1					1
	p. R143Q	1	0			1		
Amplification		20	8	1	1	2	12	4
Mutation & Amplification		4	1		1		3	

Fig 1 A) Time from diagnosis to the patient's death is indicated by the black arrow.

Q2: To complete the mutational landscape, what is the status of KRAS in selected patients, since mutations on both KRAS and EGFR driver oncogenes have been associated to chemotherapy

resistance (for instance, include PMID: 38023198, 27055085)?

Answer: We concur with you that both KRAS and EGFR mutations correlate with chemotherapy resistance. EGFR and KRAS mutations exhibit mutual exclusivity in lung adenocarcinoma, as reported in research, among elderly patients where co-occurrence prevalence is $\leq 0.52\%$, consistent with plasma ddPCR analysis of 87 EGFR-mutant cases revealing no KRAS co-mutations (Sacher et al, 2016, Liu et al, 2023). In addition, there are multiple studies supporting the mutual exclusion of EGFR and KRAS mutations (Soung et al, 2005, Do et al, 2008). Similarly, HER2 and KRAS mutations demonstrate negligible co-occurrence: analysis of 44 HER2-mutant NSCLC patients identified no KRAS co-mutations, while a cohort of 36 HER2-altered cases likewise showed no KRAS alterations (Lee et al, 2020, Ahn et al, 2023). Within the cohort of 81 HER2-altered patients identified at our institution (Table EV1), KRAS co-mutations were observed in only two cases: one in a patient harboring a HER2 G776delinsLC mutation and another in a patient with HER2 amplification. Notably, no KRAS co-mutations were detected among the seven patients carrying the HER2 S310F mutation. Given this low co-occurrence frequency of KRAS mutations with EGFR/HER2 mutations, our study focused specifically on patients harboring HER2 S310F mutations with concomitant EGFR co-mutations but not KRAS co-mutations.

Q3: Importantly, therapy-induced EGFR mutations also induce phenotypic transformation from NSCLC to SCLC. HER2 mutations have been reported in SCLC patients. Therefore, what could be the implication of this mechanism in the suggested model, can authors discard histologic alterations during acquired resistance? (Revise PMID: 34729945, 37215577).

Answer: We agree with your statement that tissue type transition from non-small cell lung cancer to small cell lung cancer is one of the important mechanisms leading to EGFR-TKI resistance. We carefully read the two studies mentioned by the reviewer, which respectively demonstrated the expression of HER2 in SCLC and the activation mechanism of EGFR in SCLC (Takahashi et al, 2021, Rubio et al, 2023). As suggested, we further assessed whether NSCLC-to-SCLC transformation occurred in our study. Histological transformation was not observed in the collected tissue specimens, including three cases harboring concurrent HER2 S310F and EGFR mutations (Fig 4F), as well as two cases with acquired resistance to EGFR-TKI (Fig 7J). Previous report indicated that concurrent EGFR/TP53/RB1 alterations confer elevated SCLC transformation risk

(Offin et al, 2019). However, within our cohort of 81 HER2-altered patients (Table EV1), concurrent TP53/RB1 alterations were observed in only two cases: one in a patient with a HER2 G778_P780dup mutation and another in a HER2-amplified patient. Notably, no concurrent TP53/RB1 alterations were detected among the seven patients harboring the HER2 S310F mutation. Therefore, our current findings do not support the occurrence of NSCLC-to-SCLC transformation in this study. However, this possibility cannot be formally excluded for HER2 S310F-mutant NSCLC due to the limitation of the number of cases. Furthermore, whether elevated 5-HT levels represent an additional resistance mechanism in patients with SCLC transformation-mediated EGFR-TKI resistance presents an intriguing scientific question warranting dedicated investigation.

***Q4:** Include in Discussion additional evidence for HTR-targeted pharmacology in cancer (for instance, include PMID 38037454). Previous groups have highlighted ferroptosis resistance implications in other cancer subtypes.*

Answer: Thank you for this suggestion and we have incorporated the following content with yellow highlight in the Discussion: Growing evidence supported targeting serotonin receptors (HTRs) as a novel anticancer strategy, with pharmacological inhibition of HTR2B suppressing tumor proliferation across multiple malignancies including gastric, colorectal, pancreatic, and hepatocellular carcinomas (Soll et al, 2010, Jiang et al, 2017, Liu et al, 2023, Tu et al, 2023). In addition, Pharmacological targeting of HTR3 inhibits proliferation in lung adenocarcinoma (Tone et al, 2020). Guided by FDA-approved library screening data and established roles of HTR-targeting agents in oncology, we prioritized HTR3 a therapeutic target for overcoming EGFR-TKI resistance, and further validated HTR3 antagonists as a promising strategy for reversing EGFR-TKI resistance.

***Q5:** Fig. EV6B shows a single results from the JASPAR prediction, without statistical values of % of targets including this motif. Other authors have shown a FOXH1 recruitment in a HTR3A-dependent manner (PMID: 39046651). Is this consistent with their own data? They could show a more detailed analysis.*

Answer: Thank you for your valuable suggestion regarding the limitations of JASPAR database predictions. To address this, we have incorporated Figure EV4G, which illustrates three predicted P65-binding sites within the MAOA promoter region identified through JASPAR analysis (*relative*

profile score threshold: 80%). In addition, we designed primers targeting three predicted P65 binding sites within the MAOA promoter region (identified via JASPAR), which were validated by ChIP-qPCR. This analysis confirmed specific binding of P65 to Site 2 of the MAOA promoter (Fig. EV4H-4I). These data suggest that diminished P65 binding at MAOA promoter Site 2 in HER2 S310F mutant cells suppresses MAOA transcription. These new data have been added to the revised version of the manuscript in yellow highlight.

G

H

I

Fig EV4. G) The P65-binding motif was predicted by JASPAR, and schematic images of the potential P65 binding sites in the MAOA promoter region are shown. H, I) ChIP-qPCR analysis of P65 binding on MAOA promoter in H1975 EV/S310F and PC9 EV/S310F cells (n = 3). Data are presented as mean \pm SD. Statistical test: Two-tailed unpaired Student's t test.

We have carefully read reviewer-cited study demonstrating that HTR3A promotes proliferation and migration in lung adenocarcinoma cells via the FOXH1 pathway and correlates with poor prognosis in NSCLC, which did not assess pharmacological targeting of HTR3 (Wu et al, 2024). Our findings are not contradictory, as we specifically focus on the relationship between HTR3 antagonists and EGFR-TKI resistance. Consistent with the observation that HTR3A associates with adverse NSCLC outcomes, we demonstrate that HTR3 antagonism reverses aumolertinib resistance. Mechanistically, HTR3 antagonists overcome EGFR-TKI resistance by promoting ferroptosis. These distinct yet convergent conclusions collectively advance understanding of HTR3 signaling in lung cancer biology.

Q6: Include p65 and its suggested genomic binding effects on MAOA in their Graphical Abstract in Fig. 8. On the contrary, NRF2 plays a crucial role in the suggested model, however this work lacks robust evidence showing a causal involvement of NRF2 by loss-of-function experiments.

Answer: Thank you for your suggestion. The graphical abstract has been revised to highlight the relationship between P65 and MAOA.

Previous studies have demonstrated that CAMKK2 modulates ferroptosis through the AMPK/NRF2 axis (Wang et al, 2022), which has been validated in our study. 5-HT regulates the CAMKK2/AMPK/NRF2 pathway to mediate resistance to ferroptosis. Studies have reported NRF2 inhibitor ML385 treatment significantly suppressed NRF2 expression, phenocopying NRF2 knockdown effects (Tao et al, 2024). Therefore, ML385 was employed to corroborate its functional role. Subsequent assessment of ferroptosis markers and ROS levels revealed that ML385 exposure reduced NRF2, xCT, and GPX4 expression while elevating lipid ROS levels (Fig S4) These data mechanistically substantiate the critical function of NRF2 in 5-HT-mediated ferroptosis resistance. These new data have been added to the revised version of the manuscript in yellow highlight.

Appendix Figure S4 NRF2-mediated ferroptosis regulation. A) The protein levels of ferroptosis biomarkers after treatment with 5-HT alone with 5-HT alone or a combination of 5-HT and ML385 for 72 hours. B) The lipid ROS levels after treatment with aumolertinib alone or a combination of aumolertinib and ML385 in H1975 S310F and PC9 S310F cells for 72 hours (n = 3). Data are presented as mean ± SD. Statistical test: one way ANOVA.

Q7: "Ferroptosis-related indicators" is a better term instead of "biomarkers, -related genes, effectors"?

Answer: Thank you for providing us with such detailed suggestions. We have made revisions based on your suggestions. As recommended, we have revised "ferroptosis-related indicators" to "ferroptosis biomarkers" throughout the manuscript with yellow highlight.

Q8: Describe in more detailed the procedure for the establishment of H1975 AR cells.

Answer: Thank you for your valuable suggestion. H1975 aumolertinib secondary resistance cells were generated from parental H1975 cells via stepwise dose escalation over 24 weeks: beginning at 10 nM in week 2 and progressively increasing to a final concentration of 10 μM aumolertinib, as previously reported (Yue et al, 2025). We have incorporated this content with yellow highlight in the Methods.

Q9: Prepare a Supplementary Material file with uncropped WB images.

Answer: Thank you for your reminder. We have included all uncropped WB images in the Source Data.

Comments from Reviewer #3

Q1: In Figure 2D, you demonstrate that HTR3 antagonists in combination with aumolertinib

synergistically inhibit cell viability. However, to rule out potential off-target effects of these small molecules, it would be valuable to perform HTR3-specific knockdown (e.g., via siRNA, shRNA, or CRISPR/Cas9). If silencing HTR3 recapitulates the combination effect seen with the pharmacological inhibitors, it will strongly support that the observed synergy truly stems from HTR3 blockade rather than other off-target activities.

Answer: We appreciate you taking the time to read our text and your insightful feedback. Our previous results demonstrated that HTR3 antagonists in combination with aumolertinib synergistically inhibit cell viability (Fig 2D-J). To exclude potential off-target effects of HTR3 antagonists, we constructed stable HTR3A-knockdown cell lines in H1975 S310F and PC9 S310F cells using shRNA lentiviral transduction. Subsequent CCK-8 assays revealed that HTR3A knockdown significantly reduced aumolertinib's IC₅₀, indicating HTR3A knockdown in combination with aumolertinib also synergistically inhibit cell viability (Fig EV2A-D). This genetic evidence confirms that the synergy between HTR3 antagonists and aumolertinib in inhibiting cell viability is mediated by on-target HTR3 blockade rather than off-target effects. These new data have been added to the revised version of the manuscript in yellow highlight.

Fig EV2 HTR3A knockdown potentiates the sensitivity to aumolertinib in HER2 S310F-mutant cells. A, B) Western blot validation of stable HTR3A knockdown cell lines. C, D) The aumolertinib IC₅₀ value of H1975 S310F and PC9 S310F cells were calculated between shCtrl and shHTR3A groups after treating with almonertinib at the indicated concentrations for 5 days.

Q2: In Figure 2B, ALK inhibitors were also among the candidate agents identified. I wonder why

ALK inhibitors were also effective in overcoming the resistance.

Answer: Thank you for your suggestion. EGFR activation and autophosphorylation of the receptor's intracellular tyrosine kinase structural domain initiate several downstream signaling pathways, such as MAPK/ERK, JAK/STAT, and PI3K/AKT (Wang et al, 2023). Key oncogenic pathways activated by ALK include JAK-STAT, MAPK/ERK, PLC γ , and PI3K-AKT (Golding et al, 2018). Therefore, crosstalk exists between the downstream signaling pathways of ALK and EGFR. ALK fusion mutations have been demonstrated to confer resistance to EGFR-TKI via bypass activation mechanisms (Liu et al, 2019, Zeng et al, 2022). ALK inhibitors retain efficacy in patients with EGFR-TKI resistance harboring concurrent ALK fusion mutations (Guo et al, 2024). In addition, MET amplification represents a recognized mechanism of acquired resistance to EGFR-TKI, driving sustained activation of downstream signaling pathways including MAPK/ERK, JAK-STAT, and PI3K-AKT (Urbanska et al, 2023). Studies have reported retained efficacy of ALK inhibitors in patients with EGFR-TKI resistance mediated by MET amplification (Urbanska et al, 2023). Similarly, HER2 aberrations activate core EGFR downstream pathways—including MAPK/ERK, JAK-STAT, and PI3K-AKT signaling—leading to constitutive oncogenic signaling. We hypothesize that, analogous to the mechanism underlying their efficacy in MET-amplified EGFR-TKI resistance, ALK inhibitors may overcome HER2 aberration-driven EGFR-TKI resistance through suppression of these shared dysregulated downstream effectors.

Q3: *You demonstrate in Figure 3E that 5-HT levels are notably elevated in resistant cells. It would be interesting to examine whether reducing 5-HT itself (e.g., via inhibition of its synthesis or by enhancing its metabolism) can restore sensitivity to aumolertinib. Such an experiment would provide direct causal evidence that 5-HT accumulation drives resistance.*

Answer: Thank you for your valuable suggestion. We have demonstrated that elevated levels of 5-HT were associated with the development of resistance to aumolertinib (Fig 3D-I). And 5-HT accumulation in HER2 S310F mutant cells is due to decreased expression of MAOA, which is an enzyme catalyzing 5-HT degradation (Fig 4). As suggested, to further examine whether reducing 5-HT can restore sensitivity to aumolertinib, we performed rescue experiments by overexpressing MAOA in H1975 S310F and PC9 S310F cells. MAOA overexpression significantly reduced extracellular 5-HT levels and decreased the IC₅₀ of Aumolertinib (Fig EV3A-F). These findings

revealed that overexpression of MAOA reduced extracellular 5-HT levels, thereby restoring sensitivity to aumolertinib.

Fig EV3 MAOA expression regulate aumolertinib resistance. A, D) H1975 S310F and PC9 S310F cells stably overexpressing MAOA were validated by Western blot. B, E) 5-HT levels in the supernatant (n = 3). C, F) The aumolertinib IC₅₀ value of H1975 S310F and PC9 S310F cells were calculated between vector and MAOA overexpression groups after treating with almonertinib at the indicated concentrations for 5 days. Data are presented as mean ± SD. Statistical test: Two-tailed unpaired Student's t test.

Q4: Your data highlight MAOA's pivotal role in modulating 5-HT levels and resistance. You show that overexpression of MAOA can reverse resistance in resistant cells. As a complementary approach, it would be informative to knock down or knock out MAOA in parental cells (e.g., H1975, PC9, H3255) to see if such depletion confers or enhances resistance. This loss-of-function experiment would strongly reinforce the model that reduced MAOA is a central driver of the resistance phenotype.

Answer: Thank you for your valuable suggestion. We found that among genes associated with 5-HT synthesis, transport, and metabolism, only the 5-HT metabolizing enzyme MAOA was significantly downregulated in HER2 S310F cells (Fig. 4). We have confirmed that overexpression of MAOA can reverse aumolertinib resistance in HER2 S310F mutant cells (Fig EV3A-F). As suggested, to further validate the role of MAOA expression in aumolertinib resistance, we established stable MAOA knockdown cell lines in wild-type H1975 and PC9 cells. The result

suggested that compared to shCtrl group, sh-MAOA significantly increased aumolertinib's IC50, indicating that reduced MAOA expression enhances resistance to aumolertinib (Fig EV3G-J). This finding robustly validated the critical role of MAOA expression in modulating aumolertinib resistance. These new data have been added to the revised version of the manuscript in yellow highlight.

Fig EV3 MAOA expression regulate aumolertinib resistance. G, I) Western blot validation of stable MAOA knockdown cell lines. H, J) The aumolertinib IC50 value of H1975 and PC9 cells were calculated between shCtrl and shMAOA groups after treating with almonertinib at the indicated concentrations for 5 days. Data are presented as mean \pm SD. Statistical test: Two-tailed unpaired Student's t test.

Q5: *Since MAOA plays key roles in various biological processes, including mood regulation and tumor biology, there may be previous studies or case reports describing MAOA mutations or altered expression in the context of cancer drug resistance. Including a brief literature overview or discussion of MAOA's known roles in drug resistance could help frame your findings in a broader context.*

Answer: Thank you for this insightful comment. We have incorporated and highlighted the following clinically relevant aspects of MAOA's role and therapeutic relevance in oncology into the Discussion: Reduced MAOA expression correlates with aggressive malignant phenotypes and poor clinical prognosis across multiple cancers including gastric, hepatocellular, and cholangiocarcinoma (Alpini et al, 2008, Pang et al, 2020, Wang et al, 2023). Conversely, elevated

MAOA expression in prostate cancer promotes tumor growth, metastasis, stemness, and therapy resistance(Li et al, 2020). Emerging evidence further implicates MAOA in modulating the tumor immune microenvironment, underscoring its multifaceted role in tumors(Wang et al, 2021, Zhao et al, 2025). Nevertheless, the relationship between MAOA and therapeutic resistance remains underexplored, with only one study reporting crosstalk between MAOA and androgen receptor expression in PCa. Genetic or pharmacologic targeting of MAOA enhanced the growth-inhibition efficacy of enzalutamide, darolutamide, and apalutamide in both androgen-dependent and castration-resistant prostate cancer cells(Wei et al, 2021).

We tried our best to improve the manuscript and made some changes highlighted in yellow in the revised paper which will not influence the content and framework of the paper. We appreciate for Editors/Reviewers' warm work earnestly, and hope the correction will meet with approval. Once again, thank you very much for your comments and suggestions.

Based on our new data and additional experiments, our results in this revised manuscript collectively reveal the the critical role of MAOA and 5-HT in aumolertinib resistance, further delineating the transcriptional regulation of MAOA in HER2 S310F-mutant cells. These data robustly validate 5-HT as a mediator of aumolertinib resistance and support the therapeutic potential of combining HTR3 antagonists with aumolertinib to reverse EGFR-TKI resistance. We have revised the manuscript based on the reviewers' advice and suggestions. Hopefully, our revision is fit for publication on *EMBO Molecular Medicine*. We are looking forward to hearing from you.

Best regards

Hua Guo

Associate professor

Tianjin Medical University, Cancer Institute and Hospital

References

- Ahn BC, Han YJ, Kim HR, Hong MH, Cho BC and Lim SM (2023). "Real World Characteristics and Clinical Outcomes of HER2-Mutant Non-Small Cell Lung Cancer Patients Detected by Next-Generation Sequencing." *Cancer Res Treat* 55(2): 488-497.
- Alpini G, Invernizzi P, Gaudio E, Venter J, Kopriva S, Bernuzzi F, Onori P, Franchitto A, Coufal M, Frampton G, et al. (2008). "Serotonin metabolism is dysregulated in cholangiocarcinoma, which has implications for tumor growth." *Cancer Res* 68(22): 9184-9193.
- Do H, Krypuy M, Mitchell PL, Fox SB and Dobrovic A (2008). "High resolution melting analysis for rapid and sensitive EGFR and KRAS mutation detection in formalin fixed paraffin embedded biopsies." *BMC Cancer* 8: 142.
- Golding B, Luu A, Jones R and Vilorio-Petit AM (2018). "The function and therapeutic targeting of anaplastic lymphoma kinase (ALK) in non-small cell lung cancer (NSCLC)." *Mol Cancer* 17(1): 52.
- Guo Y, Zhang R, Meng Y, Wang L, Zheng L and You J (2024). "Case report: Durable response of ensartinib targeting EML4-ALK fusion in osimertinib-resistant non-small cell lung cancer." *Front Pharmacol* 15: 1359403.
- Jiang SH, Li J, Dong FY, Yang JY, Liu DJ, Yang XM, Wang YH, Yang MW, Fu XL, Zhang XX, et al. (2017). "Increased Serotonin Signaling Contributes to the Warburg Effect in Pancreatic Tumor Cells Under Metabolic Stress and Promotes Growth of Pancreatic Tumors in Mice." *Gastroenterology* 153(1): 277-291.e219.
- Lee K, Jung HA, Sun JM, Lee SH, Ahn JS, Park K and Ahn MJ (2020). "Clinical Characteristics and Outcomes of Non-small Cell Lung Cancer Patients with HER2 Alterations in Korea." *Cancer Res Treat* 52(1): 292-300.
- Li J, Pu T, Yin L, Li Q, Liao CP and Wu BJ (2020). "MAOA-mediated reprogramming of stromal fibroblasts promotes prostate tumorigenesis and cancer stemness." *Oncogene* 39(16): 3305-3321.
- Liu H, Huang Q, Fan Y, Li B, Liu X and Hu C (2023). "Dissecting the novel abilities of aripiprazole: The generation of anti-colorectal cancer effects by targeting Gaq via HTR2B." *Acta Pharm Sin B* 13(8): 3400-3413.
- Liu J, Mu Z, Liu L, Li K, Jiang R, Chen P, Zhou Q, Jin M, Ma Y, Xie Y, et al. (2019). "Frequency, clinical features and differential response to therapy of concurrent ALK/EGFR alterations in

Chinese lung cancer patients." *Drug Des Devel Ther* 13: 1809-1817.

· Liu X, Jiang G, Sun X, Su G, Zhang X, Shen D and Yan N (2023). "Relationship between driver gene mutations and clinical pathological characteristics in older lung adenocarcinoma." *Front Oncol* 13: 1275575.

· Offin M, Chan JM, Tenet M, Rizvi HA, Shen R, Riely GJ, Rekhtman N, Daneshbod Y, Quintanal-Villalonga A, Penson A, et al. (2019). "Concurrent RB1 and TP53 Alterations Define a Subset of EGFR-Mutant Lung Cancers at risk for Histologic Transformation and Inferior Clinical Outcomes." *J Thorac Oncol* 14(10): 1784-1793.

· Pang YY, Li JD, Gao L, Yang X, Dang YW, Lai ZF, Liu LM, Yang J, Wu HY, He RQ, et al. (2020). "The clinical value and potential molecular mechanism of the downregulation of MAOA in hepatocellular carcinoma tissues." *Cancer Med* 9(21): 8004-8019.

· Rubio K, Romero-Olmedo AJ, Sarvari P, Swaminathan G, Ranvir VP, Rogel-Ayala DG, Cordero J, Günther S, Mehta A, Bassaly B, et al. (2023). "Non-canonical integrin signaling activates EGFR and RAS-MAPK-ERK signaling in small cell lung cancer." *Theranostics* 13(8): 2384-2407.

· Sacher AG, Paweletz C, Dahlberg SE, Alden RS, O'Connell A, Feeney N, Mach SL, Jänne PA and Oxnard GR (2016). "Prospective Validation of Rapid Plasma Genotyping for the Detection of EGFR and KRAS Mutations in Advanced Lung Cancer." *JAMA Oncol* 2(8): 1014-1022.

· Shi R, Radulovich N, Ng C, Liu N, Notsuda H, Cabanero M, Martins-Filho SN, Raghavan V, Li Q, Mer AS, et al. (2020). "Organoid Cultures as Preclinical Models of Non-Small Cell Lung Cancer." *Clin Cancer Res* 26(5): 1162-1174.

· Soll C, Jang JH, Riener MO, Moritz W, Wild PJ, Graf R and Clavien PA (2010). "Serotonin promotes tumor growth in human hepatocellular cancer." *Hepatology* 51(4): 1244-1254.

· Soung YH, Lee JW, Kim SY, Seo SH, Park WS, Nam SW, Song SY, Han JH, Park CK, Lee JY, et al. (2005). "Mutational analysis of EGFR and K-RAS genes in lung adenocarcinomas." *Virchows Arch* 446(5): 483-488.

· Takahashi K, Taki S, Yasui H, Nishinaga Y, Isobe Y, Matsui T, Shimizu M, Koike C and Sato K (2021). "HER2 targeting near-infrared photoimmunotherapy for a CDDP-resistant small-cell lung cancer." *Cancer Med* 10(24): 8808-8819.

· Tao J, Mao M, Lu Y, Deng L, Yu S, Zeng X, Jia W, Wu Z, Li C, Ma R, et al. (2024). " Δ Np63 α promotes radioresistance in esophageal squamous cell carcinoma through the PLEC-KEAP1-NRF2

feedback loop." *Cell Death Dis* 15(11): 793.

· Tone M, Tahara S, Nojima S, Motooka D, Okuzaki D and Morii E (2020). "HTR3A is correlated with unfavorable histology and promotes proliferation through ERK phosphorylation in lung adenocarcinoma." *Cancer Sci* 111(10): 3953-3961.

· Tu RH, Wu SZ, Huang ZN, Zhong Q, Ye YH, Zheng CH, Xie JW, Wang JB, Lin JX, Chen QY, et al. (2023). "Neurotransmitter Receptor HTR2B Regulates Lipid Metabolism to Inhibit Ferroptosis in Gastric Cancer." *Cancer Res* 83(23): 3868-3885.

· Urbanska EM, Grauslund M, Koffeldt PR, Truelsen SLB, Löfgren JO, Costa JC, Melchior LC, Sørensen JB and Santoni-Rugiu E (2023). "Real-World Data on Combined EGFR-TKI and Crizotinib Treatment for Acquired and De Novo MET Amplification in Patients with Metastatic EGFR-Mutated NSCLC." *Int J Mol Sci* 24(17).

· Wang C, Zhang Y, Zhang T, Xu J, Yan S, Liang B and Xing D (2023). "Epidermal growth factor receptor dual-target inhibitors as a novel therapy for cancer: A review." *Int J Biol Macromol* 253(Pt 7): 127440.

· Wang S, Yi X, Wu Z, Guo S, Dai W, Wang H, Shi Q, Zeng K, Guo W and Li C (2022). "CAMKK2 Defines Ferroptosis Sensitivity of Melanoma Cells by Regulating AMPK–NRF2 Pathway." *J Invest Dermatol* 142(1): 189-200.e188.

· Wang YC, Wang X, Yu J, Ma F, Li Z, Zhou Y, Zeng S, Ma X, Li YR, Neal A, et al. (2021). "Targeting monoamine oxidase A-regulated tumor-associated macrophage polarization for cancer immunotherapy." *Nat Commun* 12(1): 3530.

· Wang YY, Zhou YQ, Xie JX, Zhang X, Wang SC, Li Q, Hu LP, Jiang SH, Yi SQ, Xu J, et al. (2023). "MAOA suppresses the growth of gastric cancer by interacting with NDRG1 and regulating the Warburg effect through the PI3K/AKT/mTOR pathway." *Cell Oncol (Dordr)* 46(5): 1429-1444.

· Wei J, Yin L, Li J, Wang J, Pu T, Duan P, Lin TP, Gao AC and Wu BJ (2021). "Bidirectional Cross-talk between MAOA and AR Promotes Hormone-Dependent and Castration-Resistant Prostate Cancer." *Cancer Res* 81(16): 4275-4289.

· Wu Z, Li J, Zhong M, Xu Z, Yang M and Xu C (2024). "HTR3A Promotes Non-small Cell Lung Cancer Through the FOXH1/Wnt3A Signaling Pathway." *Biochem Genet*.

· Yue P, He Y, Zuo R, Gong W, Wang Y, Chen L, Luo Y, Feng Y, Gao Y, Liu Z, et al. (2025). "CCDC34 maintains stemness phenotype through β -catenin-mediated autophagy and promotes

EGFR-TKI resistance in lung adenocarcinoma." *Cancer Gene Ther* 32(1): 104-121.

· Zeng Z, Wang T, He J and Wang Y (2022). "ALK-R3HDM1 and EML4-ALK fusion as a mechanism of acquired resistance to gefitinib: A case report and literature review." *Front Oncol* 12: 1010084.

· Zhao Z, Hu Y, Li H, Lu T, He X, Ma Y, Huang M, Li M, Yang L and Shi C (2025). "Inhibition of stromal MAOA leading activation of WNT5A enhance prostate cancer immunotherapy by involving the transition of cancer-associated fibroblasts." *J Immunother Cancer* 13(3).

· Zhou Q, Meng Y, Li D, Yao L, Le J, Liu Y, Sun Y, Zeng F, Chen X and Deng G (2024). "Ferroptosis in cancer: From molecular mechanisms to therapeutic strategies." *Signal Transduct Target Ther* 9(1): 55.

2nd Jul 2025

Dear Prof. Guo,

Thank you for submitting your revised study. Referees #1 and #3 reviewed your revised manuscript and as you will see below, they are satisfied with the revisions. I will therefore be able to accept your manuscript once the following editorial concerns are addressed:

1/ Referees' concerns:

Please address the remaining minor concerns from referee #3.

2/ Manuscript text:

- Please remove the yellow highlights and only keep in track changes mode any new modification in the text.
- If Peng Chen is also a corresponding author, an email address should be added on the title page.
- Please note that all corresponding authors are required to supply an ORCID ID for their name upon submission of a revised manuscript. An ORCID identifier is currently missing for Peng Chen.
- Please carefully check your text for spelling and grammar.
- Methods:
 - o Animals: for the toxicity study, state details of authority granting ethics approval and provide reference number for approval (for all in vivo experiments). Include a statement of compliance with ethical regulations.
 - o Human samples: please provide the full statement confirming that the experiments conformed to the principles set out in the WMA Declaration of Helsinki and the Department of Health and Human Services Belmont Report. If collected and within the bounds of privacy constraints, please report on age, sex and gender or ethnicity for all study participants.
 - o Statistics: please provide a statement on inclusion/exclusion criteria, blinding and randomization
- Funding Information should be part of Acknowledgments, without separate Funding section heading. The information provided should match the information provided in the submission system (currently, the Key Project of Science & Technology Development Fund of Tianjin Education Commission for Higher Education (No. 2022ZD064), the Scientific and Technological Projects of Tianjin (24ZXZSS00050) and Tianjin Key Medical Discipline (Specialty) Construction Project (TJYXZDXK-010A) are missing in the submission system).
- Declaration of competing interest should be renamed "Disclosure and Competing Interests Statement" and be placed after Acknowledgments.
- The section "Ethics approval and consent to participate" should be part of the Methods section.
- Author contributions: CRediT has replaced the traditional author contributions section because it offers a systematic machine readable author contributions format that allows for more effective research assessment. Please remove the Authors Contributions from the manuscript and use the free text boxes beneath each contributing author's name in our system to add specific details on the author's contribution.
- References: please remove the bullet points.

3/ Figures:

- Please make sure that all figures and figure panels are referenced in the text, including Appendix figures. Currently, there is a callout for Figure 8, please check and correct.
- Please clarify in the figure legends / in the methods how Western blots were performed, in particular regarding loading controls/GAPDH (i.e. Figure 4A, Figure 5J, K). Clearly indicate whether the different bands come from a same blot.
- We note that you provided Appendix Table S1 and Appendix Table S2 as separate files; this is not needed.
- Please address the queries from our data editors in the figure legends:
 1. Please note that the legends for figure 2 is not provided in the sequential manner (legend for figure 2I is provided before legend of figure 2H). This needs to be rectified.
 2. Please note that the legends for figure EV 3 is not provided in the sequential manner. This needs to be rectified.
 3. Please define the annotated p values ****/***/**/* as well as provide the exact p-values for the same in the legend of figure 2D-F; 7B, EV1 A, B, D, E, F, G, H as appropriate.
 4. Please note that the exact p values are not provided in the legends of figures 1C, H; 2H, J; 3D-F; 5B, E, H, I; 6C, D, G, H, I; 7A, D, G; EV3 B, E.
 5. Please indicate the statistical test used for data analysis in the legend of figure 2C
 6. Please note that the box plots need to be defined in terms of minima, maxima, centre, bounds of box and whiskers, and percentile in the legends of figures 1G, 6C.
 7. Please note that information related to n is missing in the legends of figures 5G, 7B, C; EV1 A-H; EV2 C, D; EV3 C, F, H, J.
 8. Please note that the error bars are not defined in the legends of figures 1H, I; EV2 C, D.
 9. Please note that scale bar and its definition are missing for figure 5I

4/ Thank you for providing Source Data. Please make sure the labeling within the excel file is in English.

5/ Checklist:

- Please fill in the entire section "Experimental study design and statistics".
- You have filled in the subsection "Studies involving specimen and field samples", which doesn't seem to apply to your study. Please check and correct if needed.

6/ I introduced minor changes in your synopsis text, please let me know if you agree or amend as you see fit:

"HER2 S310F-mutant cells develop resistance to aumolertinib through suppression of ferroptosis mediated by 5-HT accumulation. Circulating 5-HT could serve as a predictive biomarker for aumolertinib resistance and guide the use of combination regimens involving HTR3 antagonists alongside aumolertinib.

- HER2 S310F mutation contributed to third-generation EGFR-TKI resistance in LUAD.
- Accumulation of the neurotransmitter 5-HT led to aumolertinib resistance.
- Plasma 5-HT level may be a biomarker for predicting aumolertinib resistance.
- Combining aumolertinib with HTR3 antagonists represents a promising strategy to reverse aumolertinib resistance."

Thank you for providing a nice synopsis picture; please resize it to 550 px wide x 300-600 px high and make sure the resolution is high. I have cropped a small portion to serve as a thumbnail for the table of content on our webpage (attached), please let me know if you agree, or provide an alternative image (115px x 70px).

7/ As part of the EMBO Publications transparent editorial process initiative (see our Editorial at <http://embomolmed.embopress.org/content/2/9/329>), EMBO Molecular Medicine will publish online a Review Process File (RPF) to accompany accepted manuscripts.

This file will be published in conjunction with your paper and will include the anonymous referee reports, your point-by-point response and all pertinent correspondence relating to the manuscript. Let us know whether you agree with the publication of the RPF and as here, if you want to remove or not any figures from it prior to publication.

I look forward to receiving your revised manuscript.

Yours sincerely,

Lise Roth

***** Reviewer's comments *****

Referee #1 (Remarks for Author):

The authors had addressed the comments adequately.

Referee #3 (Remarks for Author):

Thank you for thoroughly addressing my comments and providing additional experimental data. Your responses have significantly strengthened the manuscript.

Before final acceptance, please carefully address the following minor issues:

-Indicate the statistical significance and p-values in the newly added Fig. EV2C, D and Fig. EV3H, J.

-Correct the inconsistency in the spelling of "aumolertinib" (currently mixed as "aumolertinib" and "almonertinib"). Please ensure

uniform spelling throughout the manuscript.

Once these minor revisions are completed, I am pleased to recommend your manuscript for acceptance.

Manuscript ID: [EMM-2025-21570-V3]

1. Referees' Concerns

Address remaining minor concerns from Referee #3.

-Indicate the statistical significance and p-values in the newly added Fig. EV2C, D and Fig. EV3H, J.

Response: We have incorporated explicit p-values into Fig. EV2C, EV2D, EV3H, and EV3J.

-Correct the inconsistency in the spelling of "aumolertinib" (currently mixed as "aumolertinib" and "almonertinib"). Please ensure uniform spelling throughout the manuscript.

Response: We are grateful to the referee for highlighting this issue. A comprehensive review has been conducted to standardize the nomenclature of aumolertinib throughout the manuscript, figures, and supplementary materials.

25th Jul 2025

Dear Prof. Guo,

Thank you for sending your revised files. I am pleased to inform you that your manuscript is accepted for publication and is now being sent to our publisher to be included in the next available issue of EMBO Molecular Medicine.

Yours sincerely,
